# CdGAP promotes prostate cancer metastasis by regulating epithelial-to-mesenchymal transition, cell cycle progression, and apoptosis

Chahat Mehra[1,2,7], Ji-Hyun Chung[1,2,7], Yi He[1,2], Mónica Lara-Márquez[1,2], Marie-Anne Goyette [3], Nadia Boufaied[1], Véronique Barrès[4], Véronique Ouellet [4], Karl-Phillippe Guérard[1], Carine Delliaux[3], Fred Saad[4,5], Jacques Lapointe [1,6], Jean-François Côté [2,3], David P. Labbé [1,2,6✉] & Nathalie Lamarche-Vane [1,2✉]

High mortality of prostate cancer patients is primarily due to metastasis. Understanding the mechanisms controlling metastatic processes remains essential to develop novel therapies designed to prevent the progression from localized disease to metastasis. CdGAP plays important roles in the control of cell adhesion, migration, and proliferation, which are central to cancer progression. Here we show that elevated CdGAP expression is associated with early biochemical recurrence and bone metastasis in prostate cancer patients. Knockdown of CdGAP in metastatic castration-resistant prostate cancer (CRPC) PC-3 and 22Rv1 cells reduces cell motility, invasion, and proliferation while inducing apoptosis in CdGAP-depleted PC-3 cells. Conversely, overexpression of CdGAP in DU-145, 22Rv1, and LNCaP cells increases cell migration and invasion. Using global gene expression approaches, we found that CdGAP regulates the expression of genes involved in epithelial-to-mesenchymal transition, apoptosis and cell cycle progression. Subcutaneous injection of CdGAP-depleted PC-3 cells into mice shows a delayed tumor initiation and attenuated tumor growth. Orthotopic injection of CdGAP-depleted PC-3 cells reduces distant metastasic burden. Collectively, these findings support a pro-oncogenic role of CdGAP in prostate tumorigenesis and unveil CdGAP as a potential biomarker and target for prostate cancer treatments.

[1] Cancer Research Program, Research Institute of the McGill University Health Centre, Montréal, QC, Canada. [2] Department of Anatomy and Cell Biology, McGill University, Montréal, QC, Canada. [3] Institut de Recherches Cliniques de Montréal, Université de Montréal, Montréal, QC, Canada. [4] Centre de Recherche du Centre Hospitalier de l'Université de Montréal et Institut du Cancer de Montréal, Montréal, QC, Canada. [5] Department of Surgery, Université de Montréal, Montréal, QC, Canada. [6] Division of Urology, Department of Surgery, McGill University, Montréal, QC, Canada. [7] These authors contributed equally: Chahat Mehra, Ji-Hyun Chung. ✉email: david.labbe@mcgill.ca; nathalie.lamarche@mcgill.ca

Prostate cancer is the second most commonly diagnosed cancer in men[1]. While patients bearing a localized tumor display high survival rates, once the tumor advances and metastasizes current therapies are limited and ineffective[2]. Thus, understanding the molecular mechanisms underlying prostate cancer progression is a pressing unmet need and the identification of novel therapeutic targets is necessary for the treatment of this disease.

Rho GTPases are a subfamily of the large Ras superfamily of small GTPases, which have important roles in cytoskeletal remodeling, cytokinesis, cell polarity, cell motility, cell invasion, and apoptosis[3]. Rho proteins act as molecular switches cycling between active GTP-bound and inactive GDP-bound states. This GDP/GTP cycle is regulated by guanine nucleotide exchange factors (GEFs) that promote the exchange of GDP for GTP while GTPase-activating proteins (GAPs) stimulate the intrinsic GTPase activity, leading to protein inactivation[4]. Given their key roles in normal cellular processes, it is not surprising that aberrant Rho signaling is frequently implicated in human tumors[3]. However, as the frequency of activating mutations in *RHO* genes is much less than in *RAS* genes in cancer patients[5], the regulators of Rho GTPases have emerged as targets of subversion in cancer[3,6]. In particular, GAPs have been assigned tumor suppressor roles in cancer due to their ability to inactivate Rho GTPases, but recent evidence has emerged contradicting the existing dogma and implicating RhoGAPs as oncoproteins in several cancers, including breast and prostate cancers[6–10].

Cdc42 GTPase-activating protein (CdGAP, also known as ARHGAP31) is a RhoGAP specific for Rac1 and Cdc42, but not Rho[11,12]. CdGAP is highly phosphorylated on serine and threonine residues in response to growth factors and is a substrate of the extracellular signal-regulated kinase (ERK), GSK-3, and p90 ribosomal S6 kinase (RSK), mediating cross-talk between the Ras/MAP kinase pathway and Rac1 regulation[13]. Previous studies have reported gain-of-function mutations in *ARHGAP31* in patients with the rare developmental Adams–Oliver syndrome (AOS), which is characterized by aplasia cutis congenita and terminal transverse limb defects[14]. In addition, there is compelling evidence to support a pro-oncogenic role for CdGAP in cancer progression. Notably, CdGAP is a serum-inducible gene and modulates cell spreading, lamellipodia formation, focal adhesion turnover, matrix-rigidity sensing, and durotaxis—implicating a role in cytoskeletal remodeling and cellular migration[15–17]. Furthermore, the loss of CdGAP in mice severely compromised embryonic vascular development and resulted in impaired VEGF-mediated angiogenesis, one of the hallmarks of cancer[18]. Moreover, CdGAP has been implicated in the regulation of the expression of E-cadherin—loss of which is a key step of epithelial-to-mesenchymal transition (EMT)—via two different mechanisms. Firstly, the expression of CdGAP has been shown to significantly disrupt mature epithelial cell–cell contacts[19]. Secondly, CdGAP was shown to translocate to the nucleus and form a functional complex with the transcriptional factor ZEB2 to repress E-cadherin expression in breast cancer cells[7]. Importantly, CdGAP mediates transforming growth factor (TGFβ)- and ErbB2-induced breast cancer cell motility and invasion in a GAP-independent manner[8]. In vivo, loss of CdGAP in ErbB2-transformed breast cancer cells impaired tumor growth and suppressed metastasis to the lungs[7]. Consistently, high expression of CdGAP correlated with poor disease-free survival in all subtypes of breast cancer patients[7].

In this study, we sought to investigate the role of CdGAP/ARHGAP31 in prostate cancer. We first interrogated publicly available prostate cancer data sets with combined gene expression and clinical data, which demonstrated a positive association between high CdGAP expression and early biochemical recurrence (BCR) in prostate cancer patients. Knockdown of CdGAP in two human castration-resistant prostate cancer cell (CRPC) lines inhibited cell motility, invasion, and proliferation, even though higher levels of Rac1-GTP were detected in CdGAP-depleted PC-3 cells. Using global gene expression approaches, we found that CdGAP regulates the expression of genes involved in EMT but also genes involved in apoptosis and cell cycle progression. We correlated this effect with an increase in the cyclin-dependent kinase (CDK) inhibitor p21 levels, a concomitant arrest in the G1 cell-cycle phase, and an increased sensitivity of CdGAP-depleted PC-3 cells to doxorubicin-induced apoptosis. Furthermore, loss of CdGAP delayed tumor initiation, decreased tumor volume and tumor size in subcutaneous xenografts, and reduced distant metastasic burden in an orthotopic model of prostate cancer. Consistently, an elevated cytoplasmic CdGAP expression in prostate cancer cells was associated with bone metastasis in prostate cancer patients, further supporting an important role for CdGAP in prostate cancer progression. Therefore, our study revealed that CdGAP is an important regulator of prostate tumor progression and metastasis.

## Results

### Elevated levels of CdGAP expression in human prostate cancer is associated with a decreased time to disease recurrence.
One of the first indications of prostate cancer recurrence following initial response to therapy is the rise of the prostate-specific antigen (PSA) in the blood of patients defined as time to BCR. Therefore, to assess the clinical relevance of CdGAP in prostate cancer, we first determine whether CdGAP/ARHGAP31 expression is associated with BCR by analyzing publicly available datasets. In The Cancer Genome Atlas (TCGA_PRAD) dataset, when stratifying patients according to *ARHGAP31* expression by maximally selected rank statistic (Supplementary Fig. 1a), Kaplan–Meier analysis revealed that patients with high *ARHGAP31* expression trends toward a shorter time to BCR ($p = 0.053$; Fig. 1a). In the Mortensen dataset, Kaplan–Meier analysis showed that patients with high *ARHGAP31* expression had a significantly shorter time to BCR ($p = 0.0064$; Fig. 1b and Supplementary Fig. 1b). Strikingly, no BCR was observed within 5 years in patients with low *ARHGAP31* in contrast to 70% of patients with high *ARHGAP31*. Moreover, Kaplan–Meier analysis of TCGA_PRAD patients stratified based on *ARHGAP31* gain and amplification (cBioPortal, www.cbioportal.org;[20]) demonstrated shorter time to BCR in patients with altered *ARHGAP31* ($p = 0.0021$; Fig. 1c). Together, these data suggest that CdGAP is a positive modulator of prostate cancer recurrence.

### CdGAP depletion in PC-3 cells increases the levels of active Rac1.
We next sought to determine the expression of CdGAP in human prostate cancer cell lines[21]. CdGAP expression was undetectable in the androgen-dependent cell line LNCaP (Fig. 2a, b, and Supplementary Fig. 2a). Low levels of CdGAP were found in the CRPC DU-145 and 22Rv1 cell lines while high CdGAP protein and mRNA levels were detected in CRPC PC-3 cells (Fig. 2a, b, and Supplementary Fig. 2a). Consistently, *ARHGAP31* gene expression level in multiple prostate cancer cell lines obtained from the Prensner dataset[22] revealed the highest *ARHGAP31* expression in the PC-3 cell line (Supplementary Fig. 2b). Similar to CdGAP expression in human breast cancer cell lines we also found an inverse correlation between CdGAP and E-cadherin expression levels in human prostate cancer cell lines (Fig. 2a, b). In addition, we observed nuclear and cytoplasmic localization of CdGAP in PC-3 cells (Supplementary Fig. 2c) as previously reported in breast cancer cells[7,13].

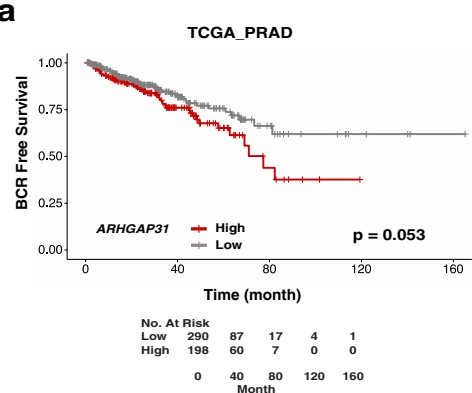

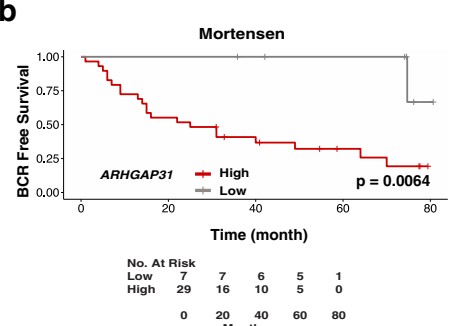

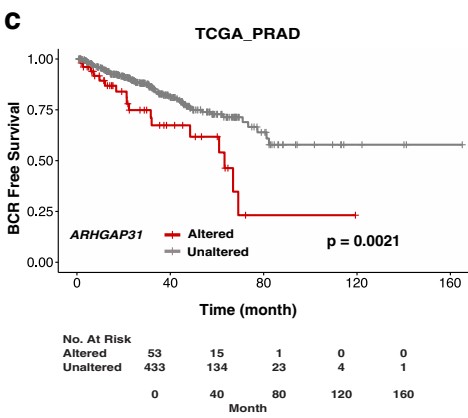

**Fig. 1 High CdGAP expression is positively correlated with cancer recurrence. a, b** Kaplan–Meier curves of biochemical recurrence (BCR) free survival for TCGA_PRAD (**a** n = 488 patients; p = 0.053) and for Mortensen et al. dataset (**b** n = 36 patients; p = 0.0064) based on *ARHGAP31* transcript levels by using maximally selected rank statistics. **c** Kaplan–Meier curves of BCR-free survival based *ARHGAP31* gene alterations (gain and amplification) in TCGA provisional data set (n = 486 patients; analyzed through cBioPortal; p = 0.0021).

To examine whether CdGAP is involved in pro-tumorigenic behaviors such as cell motility and invasion, proliferation, and tumorigenesis of CRPC cells, we generated stable PC-3 and 22Rv1 cell lines knockdown for CdGAP using short-hairpin RNA (shRNA) lentiviruses (Supplementary Fig. 2d, e). Clone 2 of shCdGAP PC-3 cells led to a 90% reduction of CdGAP protein and mRNA levels when compared with control shRNA (Fig. 2c, d, and Supplementary Fig. 2d). Similarly, a 90% reduction of CdGAP protein expression was achieved in shCdGAP 22Rv1 cells (Supplementary Fig. 2e). We assessed the effect of CdGAP depletion on the levels of active Rac1 in CdGAP-depleted PC-3 cells by performing a GST-CRIB pull-down assay. Loss of CdGAP

resulted in a 2.7-fold increase in Rac1-GTP levels (Fig. 2e), leading to significant morphological cell changes (Fig. 2f). In contrast to the elongated PC-3 control cells, CdGAP-depleted cells showed a rounded cell morphology with a decreased cell area and cell aspect ratio (Fig. 2f). Therefore, these results demonstrate that CdGAP acts as a major GAP for Rac1 in PC-3 cells.

**CdGAP silencing impairs prostate cancer cell migration, invasion, and proliferation.** To assess the role of CdGAP in prostate cancer cell migration and invasion, we performed transwell migration and invasion assays as well as wound healing assays. Control shRNA or CdGAP-depleted PC-3 and 22Rv1 cells migrated towards the bottom chamber, which contained media with 10% fetal bovine serum over a period of 24 h. Loss of CdGAP significantly impaired PC-3 and 22Rv1 cell migration and invasion. CdGAP knockdown inhibited PC-3 and 22Rv1 cell migration by 65% and 40%, respectively (Fig. 3a and Supplementary Fig. 3a), and transwell invasion through Matrigel by 74% and 40%, respectively (Fig. 3b and Supplementary Fig. 3b). Furthermore, CdGAP-depleted PC-3 cells were significantly less efficient to migrate in a wound healing assay over a period of 27 h (Fig. 3c and Supplementary Movies 1 and 2). Even though 22Rv1 cells were less migratory than PC-3 cells, loss of CdGAP in 22Rv1 cells significantly reduced the wound confluence compared to control cells (Fig. 3d and Supplementary Movies 3 and 4). We further confirmed the impact of CdGAP on human prostate cancer cell migration and invasion by ectopic expression of CdGAP in DU-145 and 22Rv1 cells and in the androgen-sensitive LNCaP cell line (Supplementary Fig. 3c). Consistently, CdGAP overexpression in all three cell lines significantly increased cell migration and invasion (Fig. 3e, f and Supplementary Fig. 3d, e). Increased migratory capacity of cells depends on their ability to rapidly attach and detach with the extracellular matrix[23]. Thus, we next determine whether CdGAP depletion also affects the ability of PC-3 cells to adhere to fibronectin and type I collagen. We found that loss of CdGAP had no significant impact on the ability of PC-3 cells to adhere to fibronectin or type 1 collagen (Fig. 3g). Then, we examined the impact of CdGAP on prostate cancer cell proliferation. CdGAP depletion significantly reduced proliferation of PC-3 and 22Rv1 cells over a period of 5 days in culture (Fig. 4a and Supplementary Fig. 3f). We extended this analysis and performed a colony formation assay that revealed a 73% decrease in the number of colonies formed by CdGAP-depleted cells compared to control shRNA PC-3 cells (Fig. 4b). Collectively, these results indicate that CdGAP is a regulator of prostate cancer cell migration, invasion, and proliferation.

**CdGAP modulates the expression of genes related to EMT, apoptosis, and cell cycle arrest.** To gain mechanistic insights into the pro-migratory and proliferative role of CdGAP in prostate cancer cells, we performed transcriptomic analysis on CdGAP-depleted PC-3 cells compared to control shRNA PC-3 cells. Differential gene expression analysis identified 1384 upregulated and 720 downregulated mRNAs in CdGAP-depleted PC-3 cells compared to control cells (Fig. 5a and Supplementary Data 1). Gene set enrichment analysis (GSEA; Hallmark) revealed that gene sets associated with EMT and apoptosis were enriched in CdGAP-depleted cells (Fig. 5b, c). In addition, gene sets associated with cell proliferation, including G2M checkpoint, E2F, and MYC targets were significantly depleted in cells with compromised CdGAP expression (Fig. 5b, c). Furthermore, gene ontology analysis centered on biological processes revealed that genes related to chemotaxis, cell motility, and the urogenital system development were amongst the most significantly affected in CdGAP-depleted cells (Fig. 5d). In this way, CdGAP has also

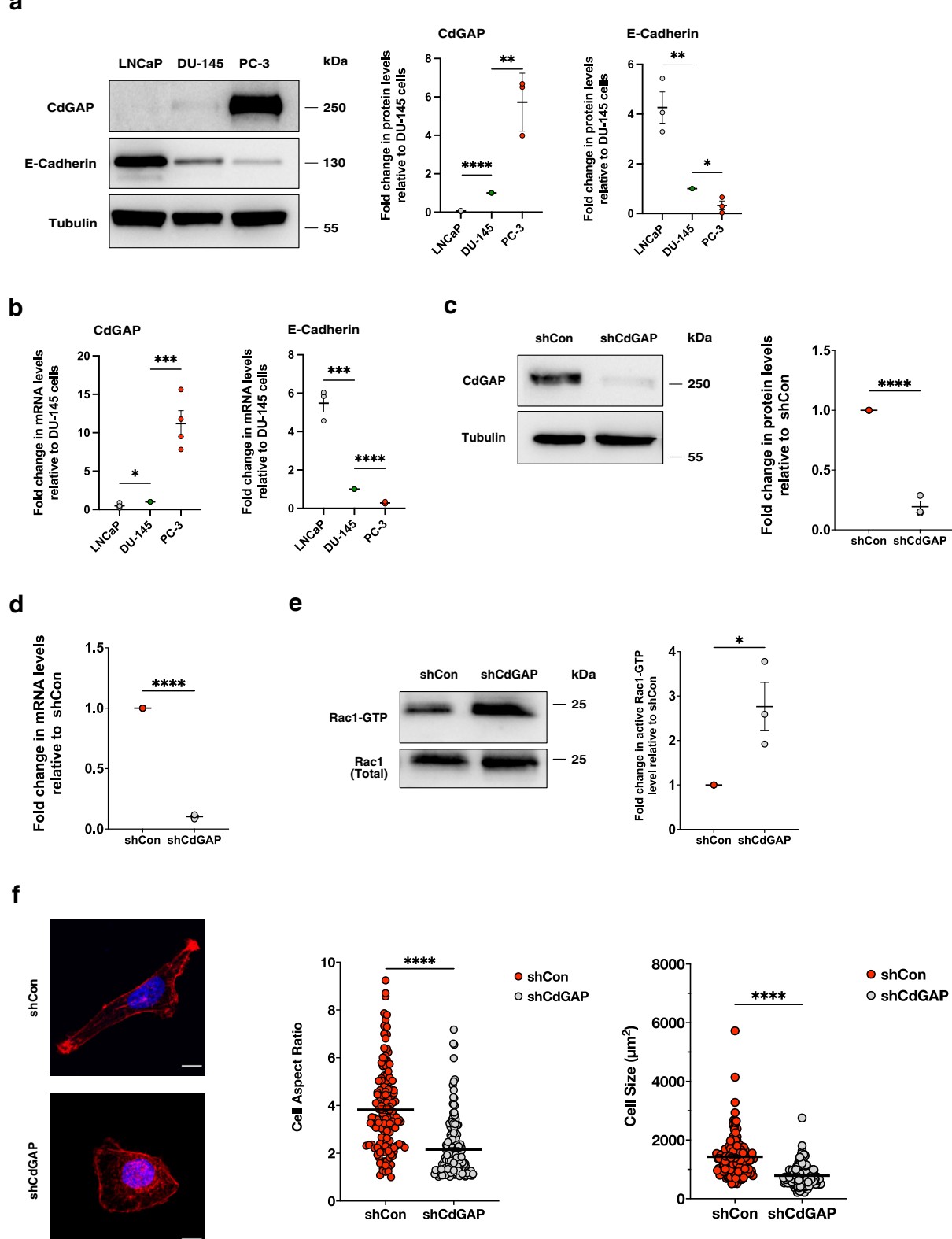

been shown to modulate EMT and cell motility gene expression profiles in breast cancer cells[7]. Indeed, loss of CdGAP in ErbB2-expressing mouse mammary tumor cells and in human breast cancer MDA-MB-231 cells resulted in a decrease of SNAIL1 and ZEB2 concomitantly with an increase of E-cadherin and reinstatement of cellular adherens junctions[7]. In contrast to the effects observed in breast cancer cells, CdGAP depletion in PC-3

cells led to a significant increase in SNAIL1 (*SNAI1*) and a decrease in E-cadherin (*CDH1*) mRNA and protein levels (Fig. 5e, f). On the other hand, the levels of two mesenchymal markers N-cadherin (*CDH2*) and Slug (*SNAI2*) were significantly decreased in CdGAP-depleted PC-3 cells (Fig. 5e, f). A decrease expression of these genes has been consistently reported as correlated with a decrease in cell motility[24]. Altogether, these results

**Fig. 2 Loss of CdGAP results in elevated Rac1-GTP levels in PC-3 cells. a** Immunoblot analysis of CdGAP and E-cadherin in human prostate cancer cell lines DU-145, LNCaP, and PC-3. Tubulin was used as a loading control. Graphs provide a densitometry analysis of CdGAP and E-cadherin protein levels represented as the fold change relative to DU-145 cells ($n = 3$). **b** mRNA levels of CdGAP ($n = 4$) and E-cadherin ($n = 3$) represented as the fold change relative to DU-145 cells. **c** Immunoblot analysis of CdGAP levels in PC-3 cells infected with scrambled control (shCon) or shRNA targeting CdGAP (shCdGAP). Tubulin was used as a loading control. The graph provides a densitometry analysis of CdGAP protein levels represented as the fold change relative to control ($n = 3$). **d** mRNA levels of CdGAP represented as the fold change relative to control ($n = 3$). **e** GTP-bound Rac1 was pulled down using GST-CRIB from control (shCon) or CdGAP-depleted PC-3 (shCdGAP) cell lysates. TCL: total cell lysates. Graphs provide a densitometry analysis of GTP-bound Rac1/total Rac1 represented as the fold change relative to control ($n = 3$). **f** Control and shCdGAP PC-3 cells were plated on coverslips coated with fibronectin. Actin filaments and nuclei were stained using phalloidin-TRITC and DAPI. Scale bar represents 10 μm. Cell aspect ratio and cell size were quantified ($n = 3$). shCon: total number of cells = 130; shCdGAP: total number of cells = 166. Two-sample unpaired Student's $t$ test for comparison between two groups with Welch's correction in (**f**). Error bars indicate SEM. ****$p < 0.0001$ ***$p < 0.001$; **$p < 0.01$; *$p < 0.05$.

suggest that CdGAP affects cell motility and EMT gene expression in prostate cancer.

In addition, a subset of genes encoding cell cycle checkpoint proteins was significantly increased in CdGAP-depleted cells compared to control shRNA cells (Fig. 6a). Accordingly, the increased levels of the CDK inhibitor p21 (*CDKN1A*) (Fig. 6a, b), which is crucial in the regulation of G1 cell cycle progression[25,26] was validated by qPCR, showing a threefold increase in CdGAP-depleted PC-3 cells compared to control shRNA cells (Fig. 6b). To assess the role of CdGAP on G1 cell cycle progression, flow cytometry analysis was conducted by staining cellular DNA with propidium iodide (PI). It revealed a significant increase of cell population in the G1 phase cell cycle (61%) in CdGAP-depleted PC-3 cells compared to control shRNA cells (53%), therefore limiting the percentage of cells in the S (from 26 to 22%) and G2 (from 21 to 16%) phases (Fig. 6c). Next, we examined whether CdGAP could affect cell death by inducing apoptosis in PC-3 cells submitted to a 12h-doxorubicin treatment followed by Annexin V/PI flow cytometry analysis. As shown in Fig. 6d, we observed a significant increase in the apoptotic cell population in CdGAP-depleted PC-3 cells (6%) compared to control cells (0.5%) when treated with vehicle. Increased concentrations of doxorubicin treatment induced cell apoptosis in a dose-dependent manner in both shRNA control cells and shCdGAP cells. However, CdGAP-depleted cells were significantly more sensitive to doxorubicin-induced cell apoptosis compared to control cells in all doxorubicin conditions tested (doxorubicin 5 μM; 37% in shCdGAP cells compared to 9% in control cells; Fig. 6d). Therefore, the loss of CdGAP resulted in G1 cell cycle arrest with a concomitant increase in cell apoptosis in PC-3 cells, which correlates with a decrease of cell proliferation observed in CdGAP-depleted cells (Fig. 4a, b). Taken together, these analyses revealed CdGAP as a key modulator of prostate cancer cell proliferation through the control of apoptosis and cell cycle genes.

**The loss of CdGAP delays subcutaneous tumor formation and attenuates tumorigenesis induced in vivo**. We next determined the role of CdGAP in tumorigenesis in vivo by injecting subcutaneously control shRNA cells or CdGAP-depleted PC-3 cells into athymic mice. The loss of CdGAP significantly delayed tumor formation with a 2.6-fold difference between the control group and the shCdGAP group of mice (Fig. 7a). In addition, 73% of the mice injected with CdGAP-depleted cells led to tumor formation compared to 100% of mice injected with control cells (Fig. 7a). Consistently, the endpoint tumors from the shCdGAP group of mice were smaller compared to the control group (Fig. 7b), which correlated with a significant twofold reduction in tumor volume and tumor weight from the shCdGAP cohort compared to control mice at 34 days post-injection (Fig. 7c, d). Together, these data demonstrate that CdGAP promotes tumorigenesis.

**CdGAP knockdown attenuates distant metastasis in an orthotopic model**. To further investigate CdGAP function in prostate cancer metastasis, we injected CdGAP-depleted PC-3 cells or control shRNA cells expressing luciferase into athymic mouse prostates. We then measured the resulting orthotopic xenograft formation and evaluated metastasis formation by bioluminescence imaging. In contrast to subcutaneous tumor formation (Fig. 7), loss of CdGAP did not significantly affect prostate tumor weight and volume at the endpoint (Fig. 8a–d). Histological analysis of primary tumors showed a typical adenocarcinoma morphology with no major differences between control and CdGAP-depleted injected mice (Fig. 8e). However, an increase in cell apoptosis as demonstrated by a significant increase in cleaved caspase-3 staining was detected in prostate tumors from mice injected with shCdGAP cells (Fig. 8f) whereas no difference in the cell proliferation marker Ki-67 or CD-31 staining was detected between control and shCdGAP tumors (Supplementary Fig. 4a, b). Local metastasis to the urogenital system, including kidneys and testes, was detected by post-mortem bioluminescence in 100% of control and shCdGAP group of mice (Fig. 8g, h; Supplementary Fig. 5). However, distant metastasis to the intestines was detected by post-mortem bioluminescence in 100% of control mice compared to 50% of CdGAP-depleted injected mice, which showed fewer lesions compared to control mice (Fig. 8g, h; Supplementary Fig. 5). Moreover, post-mortem bioluminescence in the legs and paws suggested distant metastasis to the bones in control mice, which was reduced in CdGAP-depleted injected mice (80% of control mice vs. 33% of shCdGAP mice) (Fig. 8g, h; Supplementary Fig. 5). Histological analysis of kidneys revealed tumorigenic lesions in both control and mice injected with shCdGAP cells (Fig. 8i), validating the bioluminescence images obtained post-mortem (Fig. 8g and Supplementary Fig. 5). Therefore, these results suggest a role for CdGAP in promoting prostate cancer metastasis.

**Increased levels of cytoplasmic CdGAP expression in human prostate cancer is associated with reduced bone metastasis-free survival**. Next, we examined the expression of CdGAP on a panel of radical prostatectomy specimens from 285 prostate cancer patients using the TF123 tissue microarray (TMA) series[27] (Supplementary Fig. 6 and Supplementary Table 1). Since we have previously reported nuclear and cytoplasmic localization of CdGAP in breast tumor specimens[7], the nuclear and cytoplasmic intensity of CdGAP expression was evaluated within each tissue core (Fig. 8j). Notably, CdGAP cytoplasmic intensity was significantly greater in tumor (T) tissue cores in comparison to matched benign adjacent (BA) tissue cores (average fold change = 4.781; $p = 1.2e{-}21$; Fig. 8j). In contrast, comparable CdGAP nuclear intensity was detected between T and matched BA tissue cores (average fold change = 1.123; $p = 0.013$; Fig. 8j). Kaplan–Meier analyses demonstrated that high CdGAP

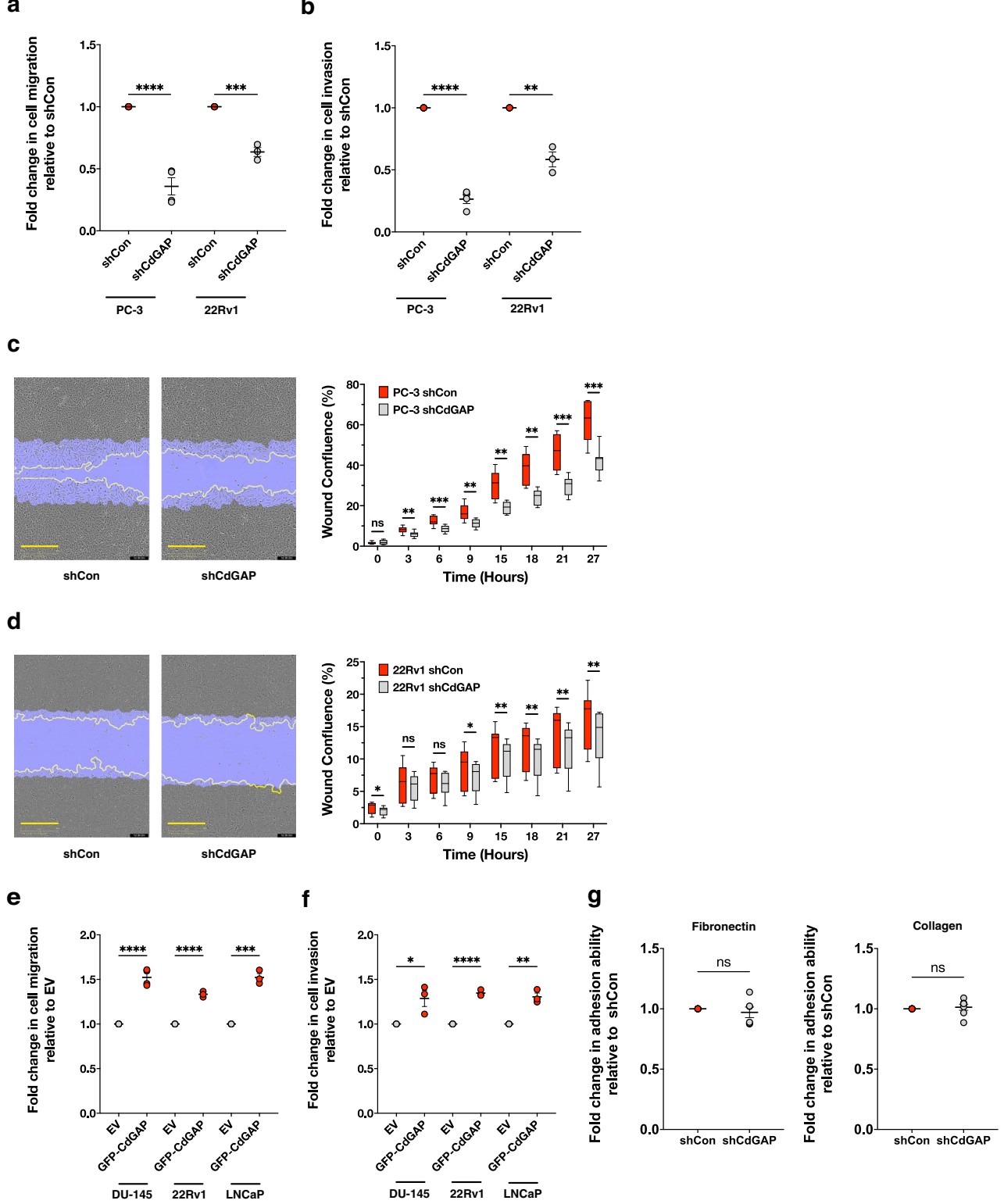

**Fig. 3 CdGAP promotes cell migration and invasion in CRPC cells. a**, **b** Quantification of transwell migration (**a**) and invasion (**b**) assays of CdGAP-depleted PC-3 and 22Rv1 (shCdGAP) cells with corresponding controls (shCon) ($n = 3$). **c**, **d** Representative images from the wound healing assays of CdGAP-depleted PC-3 (**c**) and 22Rv1 (**d**) (shCdGAP) cells with corresponding controls (shCon). Scale bar, 400 μm. Quantification of the wound confluence over a period of 27 h ($n = 3$). **e**, **f** Quantification of transwell migration (**e**) and invasion (**f**) assays of DU-145, 22Rv1, and LNCaP cells transfected with either empty vector (EV) or GFP-CdGAP (22Rv1, LNCaP: $n = 3$; DU-145: $n = 4$). **g** Adhesion assays of CdGAP-depleted PC-3 (shCdGAP) and control cells (shCon) on fibronectin and collagen type 1 ($n = 4$). Two-sample unpaired Student's $t$ test for comparison between two groups. Error bars indicate SEM. ****$p < 0.0001$ ***$p < 0.001$; **$p < 0.01$; *$p < 0.05$. n.s. not significant.

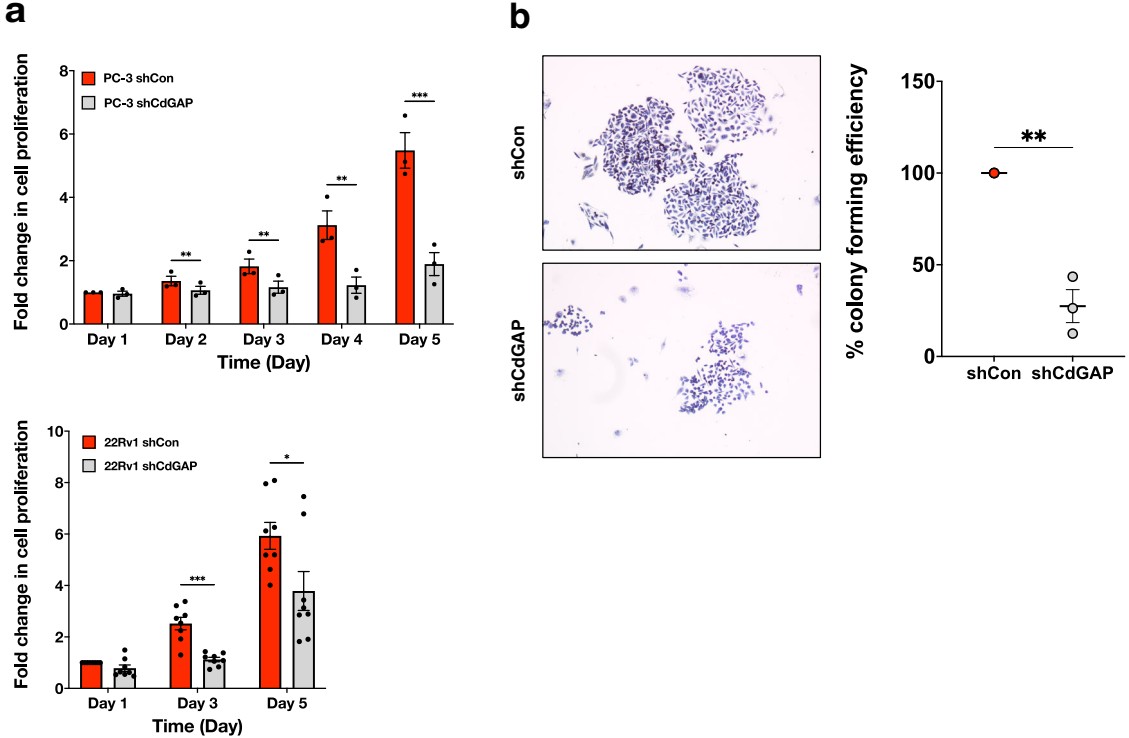

**Fig. 4 CdGAP promotes cell proliferation in CRPC cells. a** MTT assays from control (shCon) or CdGAP-depleted (shCdGAP) PC-3 and 22Rv1 cells over a period of 5 days (PC-3: $n = 3$; 22Rv1: $n = 8$). **b** Representative images of an in vitro colony formation assay. Scale bar represents 100 μm. Colony-forming efficiency is plotted relative to control PC-3 cells ($n = 3$). Two-sample unpaired Student's $t$ test for comparison between two groups (shCon; shCdGAP). Error bars indicate SEM. ***$p < 0.001$; **$p < 0.01$; *$p < 0.05$.

cytoplasmic intensity in cancer cells was associated with a trend toward increased risk of developing bone metastasis in prostate cancer patients ($p = 0.057$; Fig. 8k). Strikingly, univariable analyses revealed that patients with greater CdGAP cytoplasmic intensity in their tumor tissues (CdGAP-T) were more likely to develop bone metastatic lesions ($p = 0.005$, hazard ratio (HR) = 2.416, 95% CI: 1.310–4.453; Fig. 8l and Supplementary Table 2). Taken together, these data demonstrate the importance of CdGAP in prostate cancer metastasis and suggest that CdGAP could be used as a biomarker to identify patients at risk of progressing toward a metastatic disease.

## Discussion

Tumorigenesis is a multistep process that involves the modulation of cell proliferation, survival, migration, and invasion. The mechanisms controlling prostate cancer metastasis still remain an unresolved issue and a better understanding of prostate cancer progression will help to identify novel molecular targets for prostate cancer treatment and diagnosis. Our data presented here outline the possibility that CdGAP/ARHGAP31, a negative regulator of Rac1 and Cdc42, acts as an oncoprotein rather than a tumor suppressor in prostate cancer. We demonstrate that CdGAP is required for two CRPC cell lines, PC-3 and 22Rv1 cells, to proliferate, migrate, and invade the extracellular matrix. The mechanisms through which CdGAP promotes cell growth and migration involve the regulation of G1 cell cycle progression, apoptosis, and EMT genes (Fig. 9). Consistently, CdGAP is required for the establishment and growth of subcutaneous primary tumors. However, CdGAP expression did not affect the formation of orthotopic primary prostate tumors, highlighting the influence of the tumor microenvironment in the development of tumorigenesis[28]. Hence, CdGAP supports the development of

prostate cancer distant metastasis in an orthotopic model and is associated with bone metastasis in patients. This work has broad implications to further improve our understanding of RhoGAPs as oncogenes and their potential impact as cancer therapeutics.

Several lines of evidence suggest that CdGAP may have a pro-tumorigenic role in cancer. As a GAP for Rac1 and Cdc42, CdGAP is a key regulator of actin-cytoskeletal remodeling conferring pro-migratory roles to CdGAP[11]. Furthermore, CdGAP was shown to have a key role in the regulation of directional membrane protrusions of migrating osteosarcoma cells[16,17]. Of note, CdGAP appears to be the major RhoGAP expressed in HER2/ErbB2-induced mouse breast tumors[29]. In line with this, downstream of TGFβ and ErbB2 signaling pathways, CdGAP was shown to regulate cell migration and invasion in an ErbB2-induced mouse breast cancer cell model[8]. Furthermore, loss of CdGAP suppressed the ability of breast cancer cells to induce primary tumors and metastasize to the lungs[7]. Here, we found that elevated levels of CdGAP expression in a cohort of human prostate cancer patients were associated with an increased risk of bone metastasis in patients. These results are in good agreement with the depletion of CdGAP in PC-3 cells resulting in a reduction of distant metastasis to the intestines and potentially to the bones in an orthotopic model. In this way, analysis of gene expression datasets also revealed the positive correlation between elevated CdGAP gene expression and BCR in prostate cancer patients. Thus, this study presents data regarding *CdGAP/ARHGAP31* as a gene associated with prostate cancer metastasis and a potential target in the treatment of aggressive prostate cancer.

In order to migrate and invade, cells have to undergo a well-characterized process known as EMT. Some hallmarks of this process include upregulation of the expression of mesenchymal markers Snail1, Slug, N-cadherin, and downregulation of epithelial markers such as E-cadherin, ZO-1, and claudins. In direct

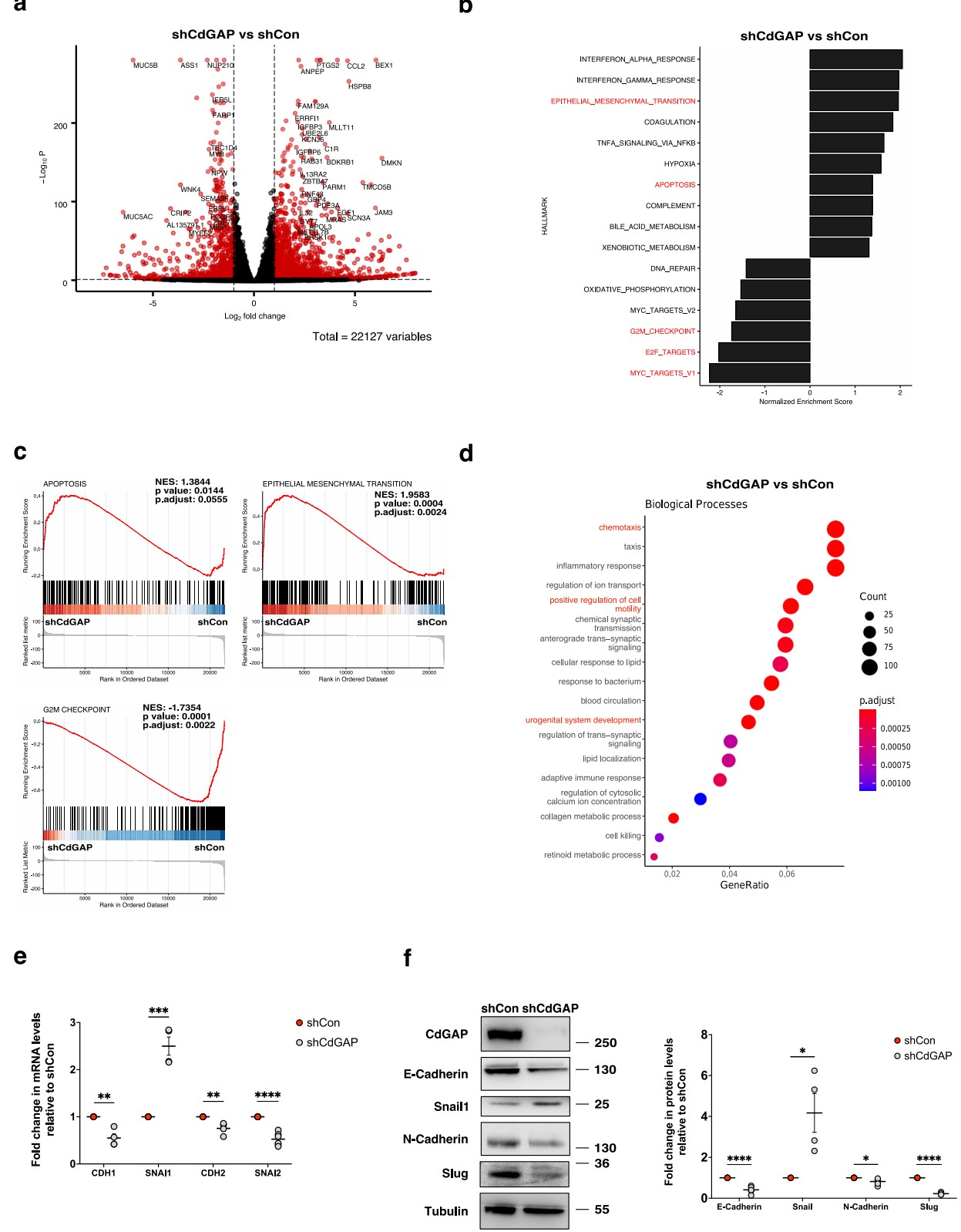

contrast to our previous study of CdGAP in breast cancer[7,8], downregulation of CdGAP resulted in a further decrease of E-cadherin levels, primarily because of the net increase in the levels of the E-cadherin transcriptional repressor Snail1. When we investigated further, we observed a decrease in other mesenchymal markers such as Slug and N-cadherin. Expression of both Slug and N-cadherin has been correlated in several reports with increased motility and an aggressive cancer phenotype[30,31]. Thus, although the marked decrease in E-cadherin levels upon CdGAP downregulation contrasts with the findings in breast cancer, the regulation of other genes hints at a differential mechanism of action of CdGAP in prostate cancer. Whether N-cadherin and

**Fig. 5 CdGAP controls a set of EMT, cell cycle, and apoptosis-related genes. a** Volcano plot of the differentially expressed genes between shCdGAP PC-3 and shControl cells. Red dots represent genes with an absolute fold change > 1 (log2FC = 1) and adjusted $p$-value < 0.01. **b** Normalized enrichment scores (NES) of significantly enriched and depleted Hallmark gene sets identified via GSEA in shCdGAP vs. shControl cells ($p$-value < 0.05). **c** Enrichment plots depicting selected gene sets significantly enriched (apoptosis, EMT) or depleted (G2M Checkpoint) in CdGAP-depleted PC-3 cells. **d** Top modulated biological processes enriched in CdGAP-depleted cells. **e** qPCR analyses of the EMT-related genes after CdGAP downregulation. (*CDH1*, *SNAI1*, *CDH2*: $n = 4$; *SNAI2*: $n = 6$). **f** Immunoblot analysis of the EMT-related proteins after CdGAP downregulation. Tubulin was used as a loading control. Graphs provide a densitometry analysis of the indicated protein levels represented as the fold change relative to control. (Snail1, Slug: $n = 4$; E-Cadherin: $n = 5$; N-Cadherin: $n = 6$). Two-sample unpaired Student's $t$ test for comparison between two groups (shCon; shCdGAP). Error bars indicate SEM. ****$p < 0.0001$ ***$p < 0.001$; **$p < 0.01$; *$p < 0.05$.

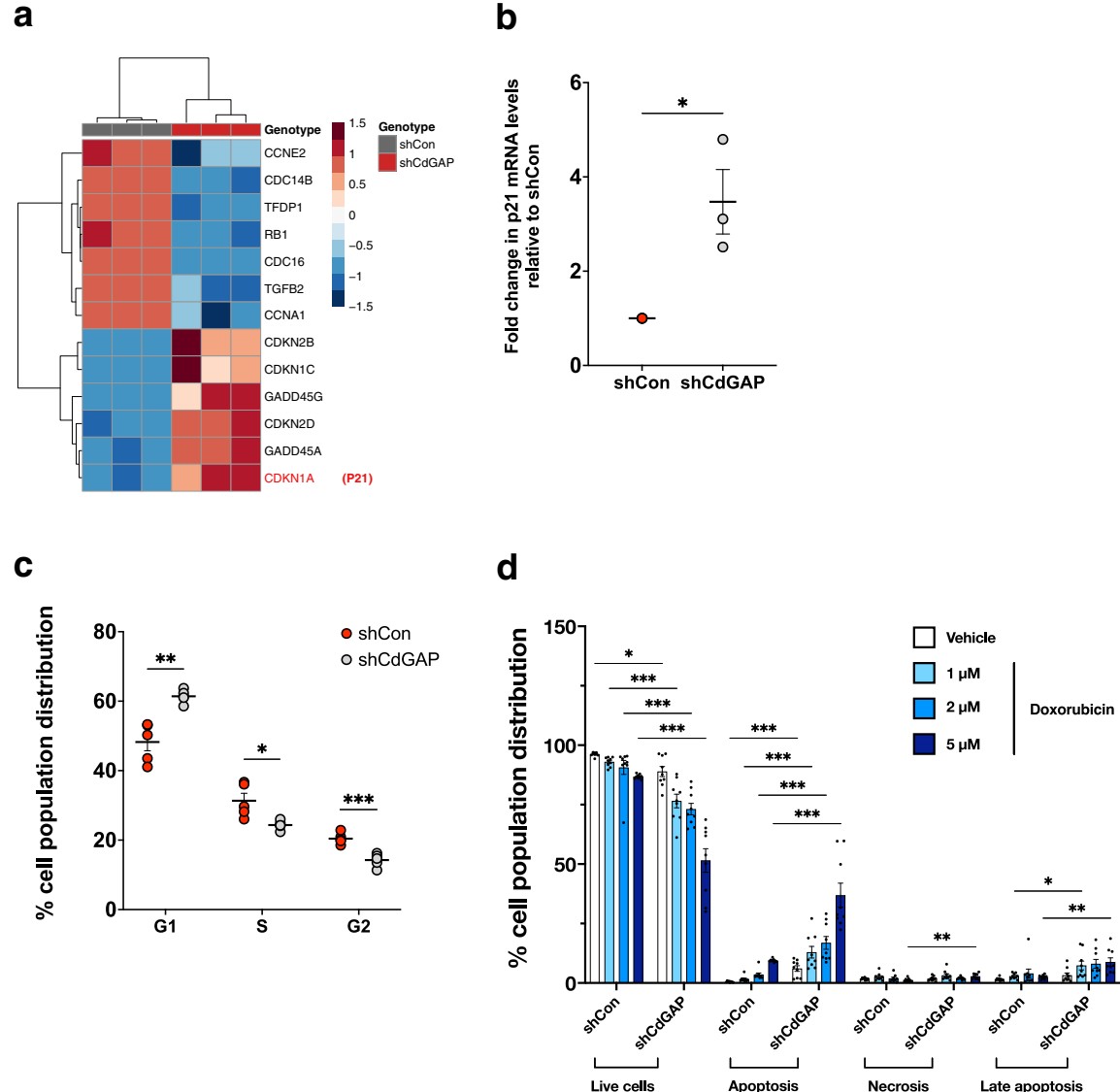

**Fig. 6 CdGAP regulates G1 cell cycle progression and apoptosis. a** Heatmap depicting cell-cycle checkpoint genes altered in CdGAP-depleted cells. **b** p21 mRNA levels in shCon (control) and shCdGAP PC-3 cells ($n = 3$). **c** Flow cytometry analysis of cell cycle distribution for CdGAP-depleted (shCdGAP) and control PC-3 cells. Cell cycle distribution is represented as the percentage of cells at each phase ($n = 3$). **d** Flow cytometry analysis of cell death in CdGAP-depleted (shCdGAP) and control PC-3 cells treated with doxorubicin (1, 2, and 5 µM) or vehicle (DMSO 0.05%) for 12 h. The percentage of cell population distribution (live, apoptosis, necrosis, late apoptosis) is represented ($n = 3$). Two-sample unpaired Student's $t$ test for comparison between two groups (shCon; shCdGAP). Error bars indicate SEM. ***$p < 0.001$; **$p < 0.01$; *$p < 0.05$.

Slug are direct targets of CdGAP during the regulation of EMT in prostate cancer need to be further investigated. Nevertheless, the differential regulation of EMT genes highlights an important role of CdGAP in the migration and invasion of prostate cancer cells.

Further investigation of the proliferative capacities using in vivo subcutaneous injections demonstrated that CdGAP-depleted tumors exhibited delayed tumor onset, reduced tumor volume, and tumor weight, in comparison to control tumors and this further substantiated the results obtained from the in vitro experiments. In contrast, prostate orthotopic injection of CdGAP-depleted cells did not alter the formation of primary tumors. These differences highlight the importance of the tumor

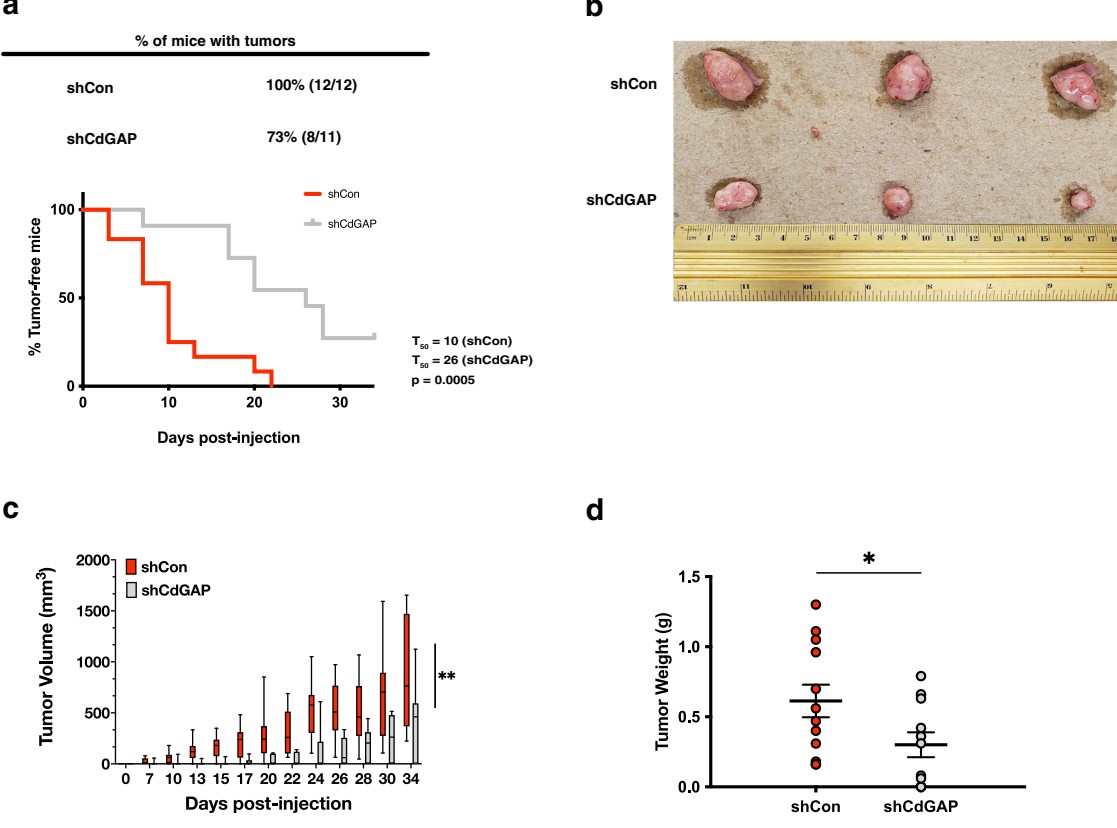

**Fig. 7 Loss of CdGAP delays subcutaneous tumor formation of PC-3 cells in vivo.** Control (shCon) or CdGAP-depleted (shCdGAP) PC-3 cells were injected into the right flanks of 7-week-old nude mice. **a** Kaplan–Meier analysis of tumor-free mice using tumor initiation as an endpoint. Time of tumor initiation was defined as when a tumor reached a volume of 20 mm$^3$. **b** Representative photographs of endpoint tumors that formed in control ($n = 12$) and shCdGAP (ndGAP) groups of mice. **c** Growth curves of subcutaneously formed tumors. Tumor volume was measured three times a week up to 34 days and is presented as the mean volume of each group (control = 12; shCdGAP = 11). Error bars indicate standard deviation (SD). **d** Tumor weight was measured at endpoint from control ($n = 12$) and shCdGAP ($n = 11$) groups of mice. Error bars indicate SEM. Two-sample unpaired Student's $t$ test for comparison between two groups (shCon; shCdGAP). **$p < 0.001$; *$p < 0.05$.

microenvironment and stroma-tumor interaction in prostate cancer growth and progression[28]. Cancer cells are sensitive to their surrounding cells and factors that contribute to reprogramming the tumor cells to either grow or arrest proliferation. The global transcriptional reprogramming in CdGAP-depleted PC-3 cells may support a positive niche for the tumors to develop in the prostate tissue environment, which may be different in a subcutaneous tumor context. For instance, the upregulation of regulatory factors including TGFβ and FGF1 in CdGAP-depleted cells could differentially influence the role of CdGAP in prostate cancer growth in a specific tumor microenvironment.

In this study, we have also observed a significant reduction in CdGAP-deficient 22Rv1 cell proliferation and a robust attenuation of cell proliferation as well as a decrease in colony-formation ability when CdGAP was depleted in PC-3 cells. The colonies in CdGAP-depleted PC-3 cells were loose and scattered from one another and unable to form compact ones as observed in control PC-3 cells. Furthermore, transcriptomics analysis of CdGAP-depleted cells revealed alterations in a subset of genes encoding cell cycle checkpoint proteins including increased levels of the CDK inhibitor p21. Consistently, we observed that the loss of CdGAP in PC-3 cells led to an arrest in the G0/G1 phase with an increase in cell apoptosis. Previous reports have implicated RhoGAPs in the regulation of CDK inhibitors[6]. Notably, depletion of ARHGAP11A in basal-like breast cancer cells was shown to lead to cell-cycle arrest mediated by p27 while depletion of

RacGAP1 led to an increase in p21 protein associated with an increase in senescence[10]. This study identified both these Rho-GAPs as oncogenic GAP essential for the regulation of cell proliferation[6,10]. By contrast, ARHGAP24 (FilGAP) emerged as a tumor-suppressor in renal cell carcinoma by inhibiting G1/S phase cell cycle progression, increasing apoptosis, and inhibited tumor growth[32]. ARHGAP10 has also been consolidated as a tumor-suppressor in ovarian cancer cells by inhibiting cell cycle progression and inducing apoptosis resulting in suppression of tumorigenesis[33].

Rho proteins organize the cytoskeleton, therefore their regulators and effectors are involved in maintaining normal homeostasis and are prone to alteration due to oncogenic transformations[3]. The pro-oncogenic role of CdGAP in breast[7,8] and prostate cancer challenges the existing paradigm and adds to the list of the emerging RhoGAPs acting as positive modulators of cancers[6]. Notably, in ovarian and colorectal cancer the expression of RacGAP1 positively correlated with lymph node metastasis and poor survival, respectively[34,35]. As well, p190A, a RhoGAP for RhoA, has been implicated as an oncogenic GAP in osteosarcoma, colorectal, lung and breast cancer[36].

In conclusion, the current study highlights the involvement of CdGAP in prostate cancer development and metastasis by regulating cell proliferation, migration, and death. CdGAP might be a valuable prognostic biomarker for metastasis and a therapeutic target in the treatment of prostate cancer.

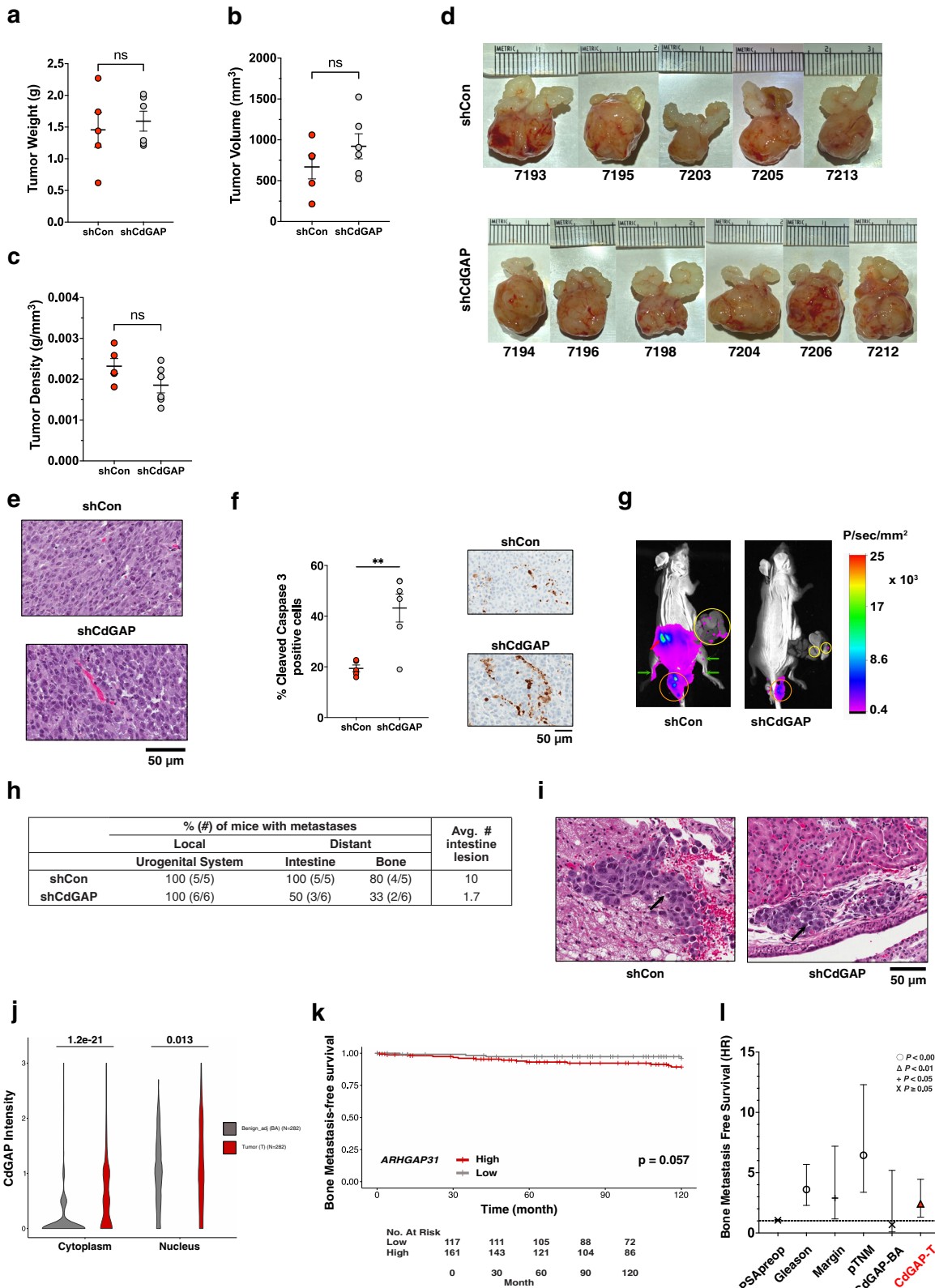

## Methods

**Mortensen and TCGA analyses**. The Mortensen dataset (GSE46602)[37] is a microarray-based dataset (Affymetrix U133 2.0 Plus), comprising 36 laser microdissected prostate cancer samples and 14 normal prostate samples. Data files with probes values (.CEL files) and sample data were downloaded from NCBI Gene Expression Omnibus (GEO) database (GEO, http://www.ncbi.nlm.nih.gov/geo/) using getGEO-function of GEOquery

package_2.54.1[38]. Read count and samples clinicopathological information from The Cancer Genome Atlas (TCGA) PRAD dataset was downloaded from the TCGA database (http://tcga-data.nci.nih.gov/tcga/)[39] using Bioconductor package TCGAbiolinks_2.14.1[40]. We used TCGA level 3 data that comprise 52 normal, 498 cancer, and 1 metastasis samples excluded from the analysis. *ARHGAP31* gene alteration (gain and amplification) information was downloaded from cBioPortal (cBioPortal, www.cbioportal.org).

**Fig. 8 CdGAP controls metastatic progression. a–d** Weight, volume, and density of primary tumors and representative photographs of primary tumors collected from control (shCon) or CdGAP-depleted (shCdGAP) PC-3 cells-injected mice at the experimental endpoint (28 days) (shCon: $n = 5$; shCdGAP: $n = 6$). Two-sample unpaired Student's $t$ test for comparison between two groups (shCon; shCdGAP). Error bars indicate SEM. ns: not significant. **e** Representative images of H&E staining of primary tumors. **f** Quantification of apoptotic cells in primary tumors by assessing the percentage of cleaved caspase-3 positive cells (shCon: $n = 5$; shCdGAP: $n = 6$). Representative images of IHC staining of cleaved caspase-3 in primary tumors. Two-sample unpaired Student's $t$ test for comparison between two groups (shCon; shCdGAP). Error bars indicate SEM. **p < 0.01. **g** Representative images of metastases found in control or CdGAP-depleted PC-3 cells-injected mice following ex vivo bioluminescent imaging at the experimental endpoint (28 days). Each mouse was exposed for 4 min after removal of the primary tumor. Yellow circle, intestine; orange circle, testis; red arrow, kidney; green arrows, legs. **h** Percentage (number; #) of mice with local and distant metastases quantified following ex vivo bioluminescent imaging at the experimental endpoint (28 days). The average number (#) of intestine lesions was quantified in each control and shCdGAP mice with metastases. **i** Representative images of H&E staining of the kidneys from control (shCon) or CdGAP-depleted (shCdGAP) PC-3 cells-injected mice. Black arrows show tumor lesions. **j** Violin plots of CdGAP intensity as scored in benign adjacent (BA) and matched tumor (T) tissue cores from the TF123 TMA ($n = 282$; cytoplasm, $p = 1.2 \times 10^{-21}$; nucleus, $p = 0.013$; Wilcoxon rank-sum test). **k** Kaplan–Meier curves of bone metastasis-free survival (10 years) based on CdGAP cytoplasmic staining in tumor cores from the TF123 TMA ($p = 0.057$). **l** Univariable analyses revealed that high CdGAP expression in tumor (CdGAP-T) cores are a prognostic factor for progression to bone metastasis (Supplementary Table 2).

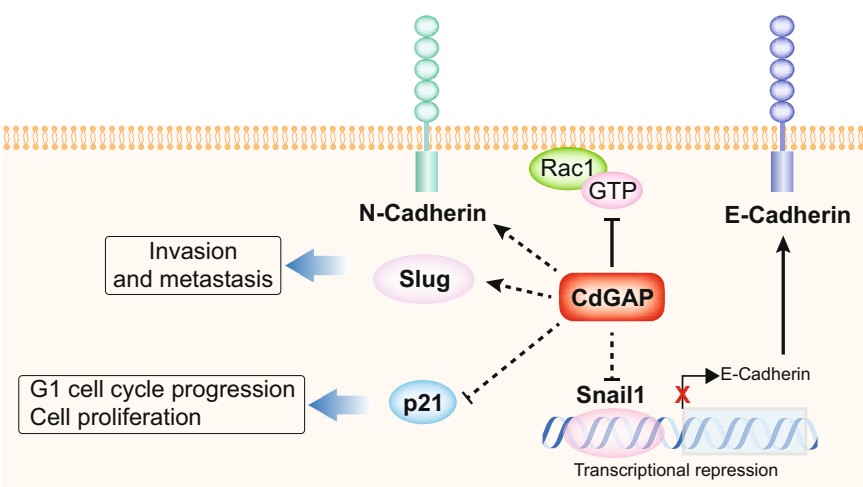

**Fig. 9 Working model for the role of CdGAP in prostate cancer metastasis.** High levels of CdGAP, a Rac1/Cdc42 inhibitor, are pro-oncogenic controlling cell invasion, metastasis, and proliferation. The mechanisms through which CdGAP promotes cell growth and migration involve the regulation of G1 cell cycle progression, apoptosis, and EMT genes. High levels of CdGAP result in increased expression of the mesenchymal markers N-cadherin and Slug, promoting invasion and metastasis while reduced levels of the CDK inhibitor p21 induce G1 cell cycle progression and cell proliferation. In addition, CdGAP negatively regulates the levels of the E-cadherin transcriptional repressor Snail1 in PC-3 cells.

*Data processing.* Microarray raw data (.CEL files) were read and preprocessed by Oligo Bioconductor package_1.50.0[41]. Probe intensities were summarized using Robust Multi Array Average (RMA) algorithm. This step includes a background correction, a quantile normalization, and a log2 transformation of the data. Probes with low intensity were filtered and the batch effect was corrected using the ComBat-function of the sva Bioconductor package_3.34.0[42]. Hugo Gene Symbols were mapped to each probe in the platform using hgu133plus2.db annotation_3.2.3 package[43] and genes with multiple probe sets were collapsed using CollapseRows-function ("MaxMean" argument) from WGCNA package_1.69[44]. TCGA RNA-seq sequencing read counts were normalized for sequencing depth using the size factor method implemented in Deseq2_1.26.0 package[45].

*Survival analyses.* To conduct survival analyses, expression data from Mortensen et al. (microarray-based) studies were transformed to z-score while expression data from TCGA (RNA-seq based) datasets was transformed using the variance-stabilizing transformation implemented in the Deseq2_1.26.0 package[45]. Patients were divided into high expression and low expression groups by optimal cutpoint calculated by survcutpoint-function of survminer_0.4.6[46] package (Supplementary Fig. 1a, b). Differences in patient's recurrence-free survival between groups were estimated by Kaplan–Meier survival analysis and log-rank tests using R package survival_3.1-12[47] and survival curves were generated using survminer_0.4.6[46] package. All data analysis and statistical tests were performed in R version 3.6.2 (2019-12-12).

**Cell culture, DNA constructs, and transfection**. PC-3, LNCaP, 22Rv1, and DU-145 prostate cancer cells (ATCC) were cultured in RPMI 1640 (Wisent: 350-000-CL) supplemented with 2 mM L-Glutamine, 10% fetal bovine serum (FBS), 1% penicillin/streptomycin, and maintained in a humidified incubator at 37 °C with

5% $CO_2$. Cell lines were regularly tested for mycoplasma contamination but they have not been authenticated. Blasticidin-resistant PC-3 cells previously transfected with empty vector pcDNA6/A (Invitrogen, Carlsbad, CA, USA) were transduced for stable bicistronic co-expression of ZsGreen and luciferase (pHIV-Luc-ZsGreen, AddGene, Watertown, MA, USA). Fluorescence-activated cell sorting was used (BD FACSAria Fusion, San Jose, CA, USA) to select the ZsGreen and luciferase positive PC-3 cells for subsequent experiments. To generate stable CdGAP-knockdown cell lines, PC-3 cells or luciferase-expressing PC-3 cells were infected with short hairpin RNA (shRNA) targeting CdGAP lentiviruses (5′-CCTCATT-TAGTTCACCTGGAACTCGAGTTCCAGGTGA ACTAAATGAGG-3′; Sigma: TRCN0000047639), and 22Rv1 cells were infected with shRNA targeting CdGAP lentiviruses (5′-CCGGCGGAGATCAGTAATTCTGGATCTCGAGATCCA GAATTACTGATCTCCGTTTTTG-3′; Sigma: TRCN0000047641) or control shRNA (Sigma: SHCON 001) purchased commercially. To select CdGAP-depleted PC-3 cells, puromycin (1 µg/ml) (Sigma: P8833) was added to the medium 48 h after infection. These cells were then plated in a 96-well plate at 1 cell/well and selected until single-cell clones were obtained. To select CdGAP-depleted 22Rv1 cells, puromycin (1 µg/ml) was added to the medium 24 h after infection. For CdGAP overexpression, DU-145, 22Rv1, and LNCaP cells were transfected with full-length pEGFPC1-mCdGAP or empty vector pEGFPC1 constructs using jetPRIME transfection reagent (Polyplus:114-07) following the manufacturer's instructions. All experiments were carried out 24 h post-transfection[13].

**Immunoblotting**. Cells were lysed in RIPA buffer containing 50 mM HEPES pH 7.5, 0.1% sodium dodecyl sulfate, 1% Triton X-100, 0.5% sodium deoxycholate, 50 mM sodium fluoride, 150 mM sodium chloride, 10 mM EDTA pH 8.0, 50 mM sodium orthovanadate, 20 mM leupeptin, 20 mM aprotinin and 1 mM

phenylmethylsulfonyl fluoride. Protein lysates were subjected to centrifugation at $10,000 \times g$ for 15 min at 4 °C to remove insoluble materials and protein concentrations were determined using the Bicinchoninic Acid Assay (BCA) protein kit (Thermo-Scientific). Equal amounts of protein samples were resolved by sodium dodecyl sulfate-polyacrylamide gel electrophoresis, transferred to nitrocellulose membranes for immunoblotting with the indicated antibodies in Supplementary Table 3, and visualized by enhanced chemiluminescence (ECL) using Clarity[TM] western ECL substrate (Bio-Rad: 1705061) and the ChemiDoc[TM] MP imaging system. All quantitative densitometry analysis on the obtained images was carried out using Image Lab software. The optical density ratios were calculated as followed: CdGAP over Tubulin; E-cadherin over Tubulin; Snail1 over Tubulin; N-cadherin over Tubulin; Slug over Tubulin; Rac1-GTP over total Rac1. The optical density fold change was calculated by normalizing the ratio of each condition to the control ratio.

**Quantitative real-time polymerase chain reaction (Q-PCR).** Total RNA was extracted using the Qiagen RNeasy kit (Qiagen: 74104). mRNA was reverse transcribed using the 5× All-In-One RT MasterMix kit (AbCAM: G485). Next, Q-PCR was performed with SYBR Green PCR Master Mix (Applied Biosystems), using primers specific to the genes of interest: CdGAP (Qiagen: QT00076671), β-actin (Qiagen: QT00095431); other primers used are listed in Supplementary Table 4. Q-PCR reactions were carried out at 95 °C for 3 min, followed by 40 cycles at 95 °C for 20 s, then at 60 °C for 30 s and finally at 72 °C for 30 min. Gene expression was normalized to β-actin RNA[7,8] and the fold change was calculated by normalizing the ratio to control cells (shCon).

**Immunofluorescence.** Cells grown on glass coverslips were fixed for 30 min in 3.7% formaldehyde in PBS before permeabilization for 5 min with 0.25% Triton-X-100 in PBS. After blocking for 30 min in a solution of PBS with 1% bovine serum albumin (BSA), coverslips were incubated overnight at 4 °C with anti-CdGAP antibodies, followed by a 45-min incubation with Alexa Fluor 488-conjugated anti-rabbit and rhodamine-conjugated phalloidin to stain for F-actin filaments. 4′,6′-diamidino-2-phenylindole (DAPI) was used to stain the nuclei. Between each step, coverslips were washed three times with PBS. Coverslips were mounted on glass slides using Prolong Gold antifade reagent (Invitrogen: P3696). Cells were examined with a motorized inverted microscope Olympus IX81 microscope using a 40× Plan-S-APO oil objective lens and images were recorded with a CoolSnap 4 K camera (Photometrics) and analyzed with Image J software[7,8]. For actin staining and quantification analyses, cells were examined with a Zeiss LSM780 confocal microscope with a 63X/1.40 oil DIC Plan-Apochromat objective and analyzed with Zen2009 and Image J software. A minimum of 30–40 cells for control or shCdGAP cells per experiment were analyzed for quantification of the cell area and aspect ratio. Aspect ratio represents the ratio of the length over the width of the cell.

**Cell migration and invasion.** For migration assays, 100,000 PC-3 (shControl; shCdGAP) or DU-145 (EV; GFP-CdGAP) cells, 50,000 22Rv1 (shControl; shCdGAP) cells, 150,000 22Rv1 (EV; GFP-CdGAP) cells, 100,000 LNCaP (EV; GFP-CdGAP) cells were resuspended in serum-free medium and seeded in the top chamber of transwell inserts (Falcon: 353097). For invasion assays, 150,000 PC-3 or DU-145 cells, 250,000 22Rv1 or LNCaP cells were plated onto a 5% Matrigel (ThermoFisher: 356234) layered over the top chamber. Cells were incubated at 37 °C overnight (PC-3, DU-145, 22Rv1 cells) or 48 h and 60 h for migration and invasion of LNCaP cells, respectively, which allow migration towards the bottom chamber containing complete medium with 10% FBS. Cells on the bottom surface of the insert were fixed in 10% formalin (BioShop: 8G56294) and stained with a crystal violet solution. Five images were taken for each transwell insert using a Nikon inverted microscope camera with a 10× objective lens (Nikon Eclipse TE300 Inverted microscope). Quantitative analysis was assessed using Image J software. Data represent the fold change relative to that of shRNA control cells or empty vector control cells obtained from at least three independent experiments[8].

**Wound-healing assays.** A 96-well IncuCyte® ImageLock microplate (Sartorius: ImageLock 4379) was coated with 1 mg/ml poly-D-lysine (Sigma: P6407-5MG) for 1 h at 37 °C. Then, wells were rinsed once with calcium and magnesium-free PBS. Totally, 15,000 PC-3 (shCon; shCdGAP) cells or 60,000 22Rv1 (shCon; shCdGAP) cells per well were seeded in triplicates, and incubated overnight. The following day, the confluency of each well was monitored. Then, IncuCyte® 96-well WoundMaker Tool (Essen Bioscience) was used to generate scratch cell monolayers, following the manufacturer's instructions. IncuCyte® S3 Live-Cell Analysis System was used for image acquisitions with a 3-h interval during a period of 27 h.

**Cell adhesion.** An in vitro adhesion assay was performed by resuspending 40,000 cells in complete media and seeding them on 96-well plates coated with 10 μg/ml type 1 collagen (BD Bioscience: 354246) or 10 μg/ml fibronectin (Sigma: F1141) for 30 min at 37 °C. Cells were fixed using 3.7% formaldehyde in PBS for 15 min, washed twice with washing buffer (0.1% BSA in RPMI), and stained with a crystal violet solution. After washing the excess dye out, the plates were allowed to dry for 1 h. Then the crystal violet stain absorbed by the cell nuclei was solubilized with 10% acetic acid and the optical density was measured at 570 nm[8].

**Cell proliferation.** To assess cell proliferation, the cell growth determination MTT (3-(4,5-Dimethylthiazol-2-yl)-2,5-diphenyltetrazolium bromide) kit (AbCAM: 211091) was used. Briefly, 250 PC-3 cells (shControl; shCdGAP) or 500 22Rv1 cells (shControl; shCdGAP) were seeded in triplicates in 96-well plates and grown over a period of five days. MTT solution was added to each well for the last 4 h of treatment on each day as per the manufacturer's protocol. Absorbance was measured at 590 nm[8]. Data represent the fold change in cell proliferation relative to that of Day 1 obtained from three independent experiments.

**Colony formation.** Two hundred and fifty cells per well in 6-well plates were resuspended in complete media for 10 days at 37 °C in a humidified incubator. On day 10, the 6-well plates were washed with PBS, fixed in 10% formalin (BioShop: 8G56294), and stained with a crystal violet solution. The excess dye was washed out with water twice and the plates were then left to dry overnight. Images were obtained with a 10× objective lens using a Nikon Eclipse TE300 Inverted microscope. Fifty cells were counted as one colony. The data represent the average of all the images per condition obtained from three independent experiments[48].

**Rac1 activation.** The CRIB domain of mouse PAK3 (amino acids 73–146) fused to glutathione S-transferase (GST-CRIB) was used to isolate GTP-bound Rac1 and was purified as described[13]. Briefly, bacterial pellets were resuspended in the lysis buffer (buffer A) containing 20 mM HEPES pH 7.5, 120 mM sodium chloride, 2 mM EDTA pH 8, 10% glycerol, and 1% Triton-X100, sonicated and centrifuged at 3000 RPM at 4 °C. Then, 30 μg of purified GST-CRIB was coupled to glutathione–agarose beads (50%) (Sigma) for 3 h at 4 °C and centrifuged at 1000 RPM for 1 min, and the pellet was washed in buffer A twice. Cell lysates (1 mg of control or shCdGAP PC-3 total cell protein) were incubated with the GST-CRIB proteins coupled to the glutathione–agarose beads for 45 min at 4 °C on a rotator. The samples were centrifuged at 1000 RPM at 4 °C for 1 min to collect the beads. After discarding the supernatant, beads were washed three times in cold RIPA buffer and resuspension in SDS sample buffer, heated at 95 °C, and then examined by immunoblotting. The levels of Rac1-GTP were assessed by densitometry and normalized to the total amount of Rac1 detected in the total cell lysates.

**Cell cycle.** Control or shCdGAP PC-3 cells were serum-starved overnight followed by a 24-hour incubation in RPMI containing 10% FBS. Totally, $1 \times 10^6$ cells were harvested, counted, and washed twice in ice-cold PBS and fixed in 70% ethanol for 1 h at 4 °C. Cells were then washed with PBS and incubated with RNase A for 1 h at 37 °C in a humidified incubator. Finally, cells were stained with 10 μg/ml PI (Sigma: P4170). Cells were subjected to flow cytometry analysis with BD FACSCanto II system. The cell cycle distribution was analyzed using the FlowJo analysis software v10.7.1 (TreeStar, Inc.)[13].

**Apoptosis.** Apoptosis was assessed in control or shCdGAP PC-3 cells using the Alexa Fluor 488 annexin V/Dead cell apoptosis kit (Invitrogen: V13241). Briefly, cells were serum-starved overnight in RPMI media supplemented with 0.25% FBS followed by a 12 h treatment with doxorubicin (1, 2, and 5 μM) (Sigma: #D1515) or DMSO 0.05% as the vehicle. Cells were subjected to flow cytometry analysis using the BD FACSCanto II system. To determine the percentage of cell population distribution, we quantified the population of apoptotic cells with fluorescence in the green emission spectrum (520 nm), necrotic cells with red fluorescence (620 nm), and late apoptotic cells with both green and red fluorescence. Data were analyzed using the FlowJo analysis software v10.7.1 (TreeStar, Inc.).

**RNA-sequencing.** RNA-sequencing was performed and analyzed as described below[7]. Briefly, total RNA from three independent samples of control shRNA PC-3 or CdGAP-depleted PC-3 (shCdGAP) cells was extracted using Qiagen RNeasy kit (Qiagen: 74104). Deep sequencing was performed using Illumina TruSeq RNA Sample Preparation Kit, Illumina TruSeq SR Cluster Kit v2, and Illumina TruSeq SBS Kit v2 (50 cycles) according to the manufacturer's procedures. Sequencing was performed at the Génome Québec Innovation Centre (McGill University) using the Illumina HiSeq 2000 platform. The quality of the raw reads was assessed with FastQC_0.11.5 and reads were aligned to the GRCh38 genome with Star 2.5.1b. Raw alignment counts were calculated with featureCounts_1.4.6 and differential expression measurements were performed with DESeq2_1.12.3. Gene ontology analyses and GSEA were conducted using ClusterProfiler_3.14.3 R package[49]. Input genes for GSEA analysis were ranked in descending order according to moderated $t$-statistic and applied to Hallmark gene sets downloaded from the Molecular Signature Database (MSigDB) using msigdbr_7.1.1R package.

**Xenograft and orthotopic injections.** To assess primary tumor growth of control or shCdGAP PC-3 cells, $1 \times 10^6$ cells were resuspended in 100 μl of serum-free RPMI containing 50% Matrigel (ThermoFisher: 356234) and injected subcutaneously using BD disposable syringe with Leur-Lok Tips (ThermoFisher: 14-823-30) into the right flanks of 7-week-old male athymic mice. Tumors were measured every 2 days with a digital caliper and the tumor volume was calculated using the following formula: $V = \pi \, (\text{length} \times \text{width}^2)/6$. After 34 days, mice were

sacrificed, and the tumors were harvested, fixed in 10% formalin (Cochiembec: F-5010Z), and subjected to analysis. Orthotopic injections of 7-week-old male athymic mice were performed as follows[50]. Briefly, male athymic mice were anesthetized and an abdominal small incision was made to expose the prostate. Totally, $2.5 \times 10^5$ control or shCdGAP PC-3-expressing luciferase cells were resuspended in 10 μl PBS with an equal volume of Matrigel and injected into the right dorsal prostate lobe. Mice were monitored daily for one week and wound clips were removed 1-week post-surgery. Tumor growth was monitored weekly thereafter via in vivo bioluminescence imaging. On the day of imaging, a 15 mg/ml luciferin solution (Perkin Elmer: #122799) was freshly prepared in PBS. Luciferin was injected intraperitoneally at a concentration of 150 mg luciferin/kg body weight. Bioluminescent imaging was performed using Bruker's in vivo Xtreme system following the manufacturer's instructions. Bioluminescence signals were normalized and presented in photons/s/mm². After 4 weeks, mice were sacrificed, and the tumors and organs potentially containing metastatic foci were dissected for formalin fixation, paraffin embedding, and tissue analysis. Ex vivo biolumi-nescent imaging at the experimental endpoint was performed on each mouse exposed for 4 min to identify the number of mice with local and distant metastasis. All animal protocols were approved by McGill University Animal Use and Care Committee, in accordance with guidelines established by the Canadian Council on Animal Care.

**Tissue microarray (TMA)**
*Construction of TMA.* The TMA TF123 series is composed of 300 radical prosta-tectomy specimens from patients participating in the Centre de recherche du Centre hospitalier de l'Université de Montréal (CRCHUM) prostate cancer bio-bank. These patients have undergone surgery at the CHUM between 1993 and 2006. For each patient, two cores (0.6 mm) of tumor (T, cancer) and two cores of BA glands were extracted from formalin-fixed paraffin-embedded radical prosta-tectomy specimens and arrayed on receiver blocks. A total of 285 prostate cancer treatment naïve specimens were used for this study (Supplementary Table 2), 15 cases were excluded due to pre-operative treatments[27].

*Scoring of CdGAP in TMA.* Using digitalized images, two different observers evaluated the nuclear frequency categorized in 0 (none), 1 (1–25%), 2 (26–75%), and 3 (76–100%), and both the nuclear and the cytoplasmic intensity (0–3 for negative, weak, moderate, high, respectively) of CdGAP within each tissue core. The average scores obtained from cores with the same histology (T or BA) were used for the statistical analyses.

*Survival analyses.* Patients were divided into two groups according to the median intensity of CdGAP in the cytoplasm. Bone metastasis-free survival was evaluated by Kaplan–Meier survival analysis and the log-rank test as described previously. Univariable Cox regression analyses were used to estimate the HRs for CdGAP using SPSS software 24.0 (SPSS Inc. Chicago, IL, USA). For univariable analyses, the serum PSA level prior to the radical prostatectomy, pathologic staging of the primary tumor (pT 2–4), Gleason Score category (6, 7 (3 + 4), 7 (4 + 3), 8 +), and margin status (negative/positive) were included in the model.

**Immunohistochemistry (IHC).** IHC was performed as described below[7]. Briefly, primary tumors were fixed in 10% formalin and paraffin-embedded. IHC was performed with Ki67 (Abcam: #ab15580; 1:300 dilution), CD31 (Abcam: # ab124432; 1:1200 dilution), and cleaved caspase-3 (Cell Signaling: #9661 s; 1:300 dilution) antibodies. All slides were counterstained using hematoxylin and eosin (H&E). Slides were scanned using a Scanscope XT digital slide scanner (Aperio, Leica Biosystems Inc., Concord, ON, Canada) and analyzed with Imagescope software (Aperio, Leica Biosystems Inc.). In human TMA staining, IHCs were performed on 4 μm-thick sections of each TMA block (n = 6) using the Benchmark XT autostainer (Ventana Medical Systems). Antigen retrieval was performed for 60 min with Cell Conditioning 1 (#950-124, Ventana Medical System, Tucson, AZ) and sections were stained using a pre-diluted (1:50) anti-CdGAP polyclonal anti-body (Sigma: HPA036380) manually added to the slides and incubated at 37 °C for 60 min. UltraView universal DAB detection kit (#760-500, Ventana Medical Sys-tem) revealed CdGAP expression, and counterstaining was achieved using hema-toxylin and bluing reagents (#760-2021 and #760-2037, Ventana Medical System). Tissues were dehydrated and mounted using SubX mounting media (Leica microsystems, Concord, ON, Canada). All sections were scanned using a VS-110 microscope with a 20 × 0.75 NA objective and a resolution of 0.3225 μm (Olympus Canada Inc., Richmond Hill, ON, Canada).

**Statistics and reproducibility.** Statistical analyses in Figs. 2–5e, f, 6–8a–c, f, Supplementary Figs. 2–4 were performed using GraphPad Prism 9 (GraphPad Software, San Diego, CA, USA). A two-sample unpaired Student's *t* test was used for comparisons between two groups. Data are presented as the mean ± SEM and a *p*-value of less than 0.05 was statistically significant. Data are representative of at least three independent experiments.

**Reporting summary.** Further information on research design is available in the Nature Research Reporting Summary linked to this article.

## Data availability
Data are available from the corresponding authors upon request. The sequencing data reported in this paper (RNA-seq) were deposited on NCBI Gene Expression Omnibus (GEO; accession number GSE160399). The source data underlying the figures can be accessed in Supplementary Data 1.

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

## Acknowledgements

We thank Dr. Aurélie Le Page for her technical assistance in performing the orthotopic injections. We thank Mr. Mathieu Simard and Dr. Aurore Dodelet-Devillers at the Small Animal Imaging Labs (SAIL) of the Research Institute of the McGill University Health Centre (RI-MUHC) for their assistance and help with the bioluminescence analyses. We thank the Immunophenotyping platform of the RI-MUHC and the Institut de Recherches cliniques de Montréal Molecular Biology and Functional Genomics core facility for assistance with the RNA-seq analyses. We thank Dr. Noriko Uetani for the drawing of the working model in Fig. 9. We are grateful to all patients who agreed to participate into the Centre de Recherche du Centre Hospitalier de l'Université de Montréal (CRCHUM) Prostate cancer tumor bank. Biobanking was done in collaboration with the Réseau de Recherche sur le cancer of the Fonds de Recherche Québec—Santé (FRQS) that is affiliated with the Canadian Tumor Repository Network (CTRNet). TMA construction was supported by the Terry Fox Research Institute. C.M. and J.-H.C. were supported by studentships from the RI-MUHC and M.L.M. from CONACyT. M.A.G. holds a Canadian Institutes of Health Research (CIHR) doctoral studentship. F.S. holds the Raymond Garneau Chair in Prostate Cancer Research—Université de Montréal. V.B., V.O., and F.S. are researchers of the CRCHUM which receives support from the FRQS. J.-.F.C. holds the TRANSAT Chair in breast cancer research. D.P.L is a Lewis Katz—Young Investigator of the Prostate Cancer Foundation and is also a Research Scholar—Junior 1 from FRQS. This research was supported by a Cancer Research Society grant to J.-F.C. and CIHR project grants (PJT-162246) to D.P.L and (PJT-153151) to N.L.V.

## Author contributions

Conception and design: C.M., J.-H.C., D.P.L., and N.L.-V. Acquisition of data: C.M., J.-H.C., Y.H., M.L.M., M.A.G., N.B., V.B., V.O., C.D., and F.S. Analysis and interpretation of data: C.M., J.-H.C., Y.H., M.L.M., M.A.G., N.B., V.O., K.P.G., J.F., D.P.L., and N.L.-V. Writing, review, and/or revision of the paper: C.M., J.-H.C., N.B., V.O., K.P.G., J.L., J.F.C., D.P.L., and N.L.-V.

## Competing interests

The authors declare no competing interests.
