## [Peer Review File · Communications Biology]

Reviewers' comments:

Reviewer #1 (Remarks to the Author):

In the manuscript entitled "The Rac1/Cdc42 regulator CdGAP promotes prostate cancer metastasis by regulating epithelial-to-mesenchymal transition, cell cycle progression, and apoptosis" the authors report elevated expression of CdGAP protein and mRNA levels in two human prostate cancer cell lines, PC-3 and DU-145 that are androgen-independent. Using global gene expression approaches, they found that CdGAP regulates the expression of EMT related genes, as well as involved in apoptosis and cell cycle progression. All three processes are fundamentally critical to tumor development progression to metastasis and the authors conclude that the master regulator maybe be of major therapeutic as well as biomarker value in prostate cancer progression. The findings are novel but their significance in the clinical progression of advanced disease was not functionally established. The major limitation of this work is the use of two prostate cancer cell lines in an orthotopic model, that do not capture the clinical setting of advanced prostate cancer, does not provide convincing data as to the functional contribution of this new player CdGAP in prostate cancer progression to metastasis. Did the authors examine the role of this protein/gene expression in human castration resistant metastatic prostate cancer cells (mCRPC) such as the 22Rv1.

It would also be relevant if consideration was given to hormone sensitive metastatic prostate cancer cell model such as the VCaP or LNCaP.

I also have some minor issues as follows:

1) There is extensive description of "Materials and Methods" with massive details given for standard assays/techniques. The Materials and Methods, "Figure Legends" and the "Results" sections need to be shortened for clarity and flow.

2) On Figure 5, panel (d) is not clear as to how the authors temporally evaluated the various modes of cell death (apoptosis, Late apoptosis, necrosis). Is there a statistically significant difference between the subpopulations entering the various cell death modes? Also not clear what the mechanistic significance of this figure is.

3) Was anoikis assessed in the context of increased migration and invasion?

Reviewer #2 (Remarks to the Author):

The manuscript of Mehra et al evaluate the RhoGAP CdGAP (GTP-ase activation protein also known as ARHGAP31) in prostate cancer as possible regulator of prostate cancer migration, invasion and proliferation, and, as potential marker for metastasis, that it has been shown to regulate breast cancer metastasis formation in the lung and it correlates with poor disease-free survival in breast cancer patients.

The authors show that CdGAP has a role also in prostate cancer using both clinical data, and a prostate cancer cell line, PC3, in vitro and in vivo. Their claim that are better supported by others. For example, in figure 3 to show that CdGAP promotes the invasion, migration and proliferation, the assays chosen are weak and could be implemented by 3D assay/presence of ECM with endothelial cells (component of the vasculature). Also, the microscopy analysis may be more complete and informative.

Although not completely novel, concerning the role of RhoGAP CdGAP in cancer metastasis, the findings of this work are novel for PROSTATE cancer metastasis field, and more widely, important for RhoGTPases in the context of cancer and cancer migration and invasion in the process of metastasis formation.

Concerning the statistical analysis and the transparency of data plotted, I strongly suggest to revise this part. 1) I cannot find the number of replicates (n) in the figure legends (only for 5 panels), nor the technical replicates. 2) Statistical analysis in M&M does not mention how it was performed the stat in graph with more than 2 columns as in fig 4e for example. 3) it would be helpful to see all the data plotted (n) (in column graphs) as the authors used the GraphPad prism tool that allow this type of graphs.

Here my comments (minor and major) on the manuscript of Mehra et al.:

Fig 2 in some data there is some variability, it would be helpful to add every n /replicate data point (not only in this figure)

Fig 2 I cannot find the n of replicates

Fig 2 a why the quantification of the expression of the 2 proteins is against DU-145, instead of using the tubulin to quantify the expression of CdGAP. The difference is obvious, but I find to use one of the 3 cell lines for the quantification not right.

Fig 2 b same here, why you do not show data relative to beta-actin (control used for real time-M&M)?

Suppl Fig 2b please add nuclear staining used in the legend as for the other stainings.

Suppl Fig 2c why did you decided to keep only one sh? As the clones 1 and 2 are the one that work best, why you did not carry the other experiments with 2 shCdGap but only with one?

Fig 2 e KD in the blot missing. Also, for the total, GAPDH or tubulin blot is missing.

Fig 2f it would be helpful to add a nuclear staining as in the suppl fig2 b.

Why in panel f. the CdGAP-depleted pc3 cells are rounded while in the supplementary fig 2 b they are all elongated? Is the round phenotype rare? Did you quantified it?

Fig 3 a how did you quantified the migration and invasion. Maybe in percentage is a better expression of the rate of migration/invasion. To couple Fig 3a top with Fig 3b and the same for the invasion may result more clear to understand to the reader.

Fig 3 a-c add PC-3 cells in Y axis as was done for DU-145 in Fig 3d

It is confusing to have data relative to fig 3 in supplementary fig 2 (d,e,f)

Supple Fig 2 e and f, it is not clear why it was assessed the adhesion to collagen and fibronectin. these are cell lines that adapt to plastic from passage one, and no difference of `adhesion` growth on plastic or ECM coated well is unsurprising, also when silencing CdGAP, that if was critical in adhesion would have had an effect also on plastic. The assay/s that would be more interesting would be the differences in phenotype, shape/protrusions/migration etc.

Fig 3 g The MTT assays measures the enzymatic activity rate. Proliferation assay by flow cytometry?

If would be helpful to have the blot for CdGAP performed at 2,3,4,5 day post in parallel with the MTT assay.

Line 167 rather than promoter, regulator, at this point of the story.

Line 231 delete migration and invasion as data in fig 5 do not show that.

Fig 4 it does not look you gained mechanistic inside with the data in this figure, because you do not show a list of possible genes affected resulting by the analisis in fig a-d, but you studied genes known to be involved in breast cancer model. Furthermore, silencing CdGAP you do not know if these effect on Snail/slug are direct or indirect.

Fig 6b The sentence in the legend is not clear. Are these tumour from 3 animals? What about the others?

In fig 6c are the data average of all 12 and 11 mice?

Fig 7 J is very hard to understand, panel I too.

CdGAP-depleted pc3 cells:

Adhesion to plate: no difference (plastic/fn/collagen) (30 minutes)

MTT enzymatic activity for cell proliferation: decreases (5 days, time course/days)

COLONY formation, cell survival test/n divisions: decreases (10 days)

Ki67 nuclear stain for proliferation, no effect (mice inj with PC3)

How do you explain no proliferation in vivo?

We would like to thank the reviewers for their helpful comments. We will now describe point-by-point how we have successfully addressed each issue.

Reviewer 1

Major Issue: Did the authors examine the the role of this protein/gene expression in human castration resistant metastatic prostate cancer cells (mCRPC) such as the 22Rv1. It would also be relevant if consideration was given to hormone sensitive metastatic prostate cancer cell model such as the VCaP or LNCaP.

Response: In addition to CdGAP-depleted PC-3 cell line, we have now generated a stable 22Rv1 cell line knockdown for CdGAP using short-hairpin RNA (shRNA) lentiviruses (**Supplementary Fig. 2e**). We have also found that CdGAP silencing in 22Rv1 cells impairs prostate cancer cell migration, invasion, and proliferation (**Fig 3a,b,d, Fig. 4a, Supplementary Fig. 3a,b,f**). We have also assessed the role of CdGAP in prostate cancer cell migration in wound healing assays and found that CdGAP-depleted PC-3 cells were significantly less efficient to migrate in a wound-healing assay over a period of 27 hours (**Fig. 3c and Supplementary Movie 1,2**). Even though 22Rv1 cells were less migratory than PC-3 cells, loss of CdGAP in 22Rv1 cells significantly reduced the wound confluence compared to control cells (**Fig. 3d and Supplementary Movie 3,4**). In addition, we have further confirmed the impact of CdGAP on human prostate cancer cell migration and invasion by ectopic expression of CdGAP in DU-145 and 22Rv1 cells and in the androgen-sensitive LNCaP cell line (**Supplementary Fig. 3e**). Consistently, CdGAP overexpression in all three cell lines significantly increased cell migration and invasion (**Fig. 3e, f and Supplementary Fig. 3d,e**).

Minor issues

- 1) There is extensive description of "Materials and Methods" with massive details given for standard assays/techniques. The Materials and Methods", "Figure Legends" and the "Results" sections need to be shortened for clarity and flow.

Response: We have revised certain parts of the Methods, Results, and Figure legends sections for clarity and flow.

- 2) On Figure 5, panel (d) is not clear as to how the authors temporally evaluated the various modes of cell death (apoptosis, late apoptosis, necrosis). Is there a statistically significant difference between the subpopulations entering the various cell death modes? Also not clear what the mechanistic significance of this figure is.

Response: Fig. 5d is now Fig. 6d. We have clarified the evaluation of the various modes of cell death in the Methods section, as followed on lines 547-551: "To determine the percentage of cell population distribution, we quantified the population of apoptotic cells with fluorescence in the

green emission spectrum (520 nm), necrotic cells with red fluorescence (620 nm), and late apoptotic cells with both green and red fluorescence. Data were analyzed using the FlowJo analysis software v10.7.1 (TreeStar, Inc.)". We have also presented the data in a different manner to show the statistical differences between control and CdGAP-depleted cells in each subpopulation.

3) Was anoikis assessed in the context of increased migration and invasion?

Response: No. Indeed, loss of CdGAP led to decrease cell migration, invasion, and proliferation in CRPC cells (Fig. 3,4), while inducing apoptosis in PC-3 cells (Fig.6). As well, orthotopic xenograft formation showed an increase in cell apoptosis in prostate tumors from mice injected with shCdGAP cells (Fig. 8f). Whether CdGAP also regulates anoikis in cancer progression remains to be investigated and will be of great interest in our future studies.

Reviewer 2

Comments: In figure 3 to show that CdGAP promotes the invasion, migration and proliferation, the assay chosen are weak and could be implemented by 3D assay/presence of ECM with endothelial cells (component of the vasculature). Also, the microscopy analysis may be more complete and informative.

Response: Indeed, a 3D co-culture model of CRPC cells and human endothelial cells to mimic interactions between prostate cancer cells and endothelial cells would be of great interest to develop and will be the focus of our future studies. In this paper, the *in vivo* xenograft studies from subcutaneous and orthotopic injections, are complementing the *in vitro* cell studies to illustrate the contribution of CdGAP to prostate tumorigenesis and metastasis recapitulating, to some extent, the complexity and heterogeneity of the tumor microenvironment. In addition, in Fig. 2f, we have replaced the actin staining images with confocal microscopy images of control and shCdGAP PC-3 cells and we have quantified the cell area and cell aspect ratio, showing a significant decreased cell area with a rounded cell morphology in CdGAP-depleted PC-3 cells.

Comments: Concerning the statistical analysis and the transparency of data plotted, I strongly suggest to revise this part. 1) I cannot find the number of replicates (n) in the figure legends (only for 5 panels), nor the technical replicates. 2) Statistical analysis in M&M does not mention how it was performed the stat in graph with more than 2 columns as in fig 4e for example. 3) it would be helpful to see all the data plotted (n) (in column graphs) as the authors used the GraphPad prism tool that allow this type of graphs.

Response: 1) We have now added the number of replicates (n) in all figure legends. 2) We have clarified the statistical analysis in the Methods and in all figure legends, as follows in Figs. 2, 3, 4, 5e,f, 6, 7, 8a-c, 8f, Supplementary Figs. 2-4: "A two-sample unpaired Student's t-test was used for comparisons between two groups (shCon;shCdGAP)." 3) We are now showing graphs with all the data plotted (n).

Other minor and major comments:

Comments: Fig 2 in some data there is some variability, it would be helpful to add every n /replicate data point (not only in this figure). Fig 2 I cannot find the n of replicates.

Response: We have added the "n" in all figure legends and plotted the graphs with all the data (n).

Comments: Fig 2 a why the quantification of the expression of the 2 proteins is against DU-145, instead of using the tubulin to quantify the expression of CdGAP. The difference is obvious, but I find to use one of the 3 cell lines for the quantification not right.

Response: Quantification of the proteins was done with tubulin and this was clarified in the figure legends and in the Methods section, as follows on lines 432-435: “The optical density ratios were calculated as follows: CdGAP over tubulin; E-cadherin over tubulin; snail1 over tubulin; n-cadherin over tubulin; slug over tubulin; Rac1-GTP over total Rac1. The optical density fold change was calculated by normalizing the ratio of each condition to control ratio.”

Comments: Fig 2 b same here, why you do not show data relative to beta-actin (control used for real time-M&M)?

Response: This issue was clarified in the Figure legends and in the Methods sections, as follows on lines 444-446: “Gene expression was normalized to β -actin RNA and the fold change was calculated by normalizing the ratio to control cells (shCon)”.

Comments: Suppl Fig 2b please add nuclear staining used in the legend as for the other stainings.

Response: Suppl. Fig. 2b is now Suppl. Fig. 2c. The figure legend has been modified accordingly.

Comments: Suppl Fig 2c why did you decided to keep only one sh? As the clones 1 and 2 are the one that work best, why you did not carry the other experiments with 2 shCdGAP but only with one?

Response: Suppl. Fig. 2c is now Suppl. Fig. 2d. Both clones were showing high downregulation of CdGAP expression, and we chose to carry on the experiments with clone 2, showing 90% reduction in CdGAP expression.

Comments: Fig 2 e KD in the blot missing. Also, for the total, GAPDH or tubulin blot is missing.

Response: In Fig. 2e, we are showing the western blot of **total Rac1** in the protein lysates, therefore we don't need to add a western blot of tubulin. We have revised the figure accordingly to better identify the lane of total Rac1. We don't do a CdGAP western blot on the same SDS-PAGE as for Rac1, since the 250KDa MW of CdGAP requires a 7.5% SDS-PAGE whereas the 20KDa MW of Rac1 requires a 12% SDS-PAGE. The CdGAP western blot was performed at the same time and is representative of Fig. 2c.

Comments: Fig 2f it would be helpful to add a nuclear staining as in the suppl fig2 b.

Response: The figure has been modified accordingly.

Comments: Why in panel f. the CdGAP-depleted pc3 cells are rounded while in the supplementary fig 2 b they are all elongated? Is the round phenotype rare? Did you quantified it?

Response: Supplementary Fig. 2b is now Suppl. Fig2c. In this figure, we are showing the localization of CdGAP in parental PC-3 cells, which are elongated cells similar to the shControl PC-3 cells in Fig. 2f. In Fig. 2f, we have replaced the actin staining images with confocal microscopy images of control and shCdGAP PC-3 cells. We have quantified the cell area and cell aspect ratio, showing a rounded cell morphology with a decreased cell area and cell aspect ratio in CdGAP-depleted PC-3 cells.

Comments: Fig 3 a how did you quantified the migration and invasion. Maybe in percentage is a better expression of the rate of migration/invasion. To couple Fig 3a top with Fig 3b and the same for the invasion may result more clear to understand to the reader.

Response: Fig. 3a and 3b have been modified and are now including results from the 22Rv1 cells knockdown for CdGAP. We have modified the figure for clarity and the images are now in Supp. Fig. 3a and 3b. We have clarified the Methods section, as follows on lines 475-479: “Five images were taken for each transwell insert using a Nikon inverted microscope camera with a 10X objective lens (Nikon Eclipse TE300 Inverted microscope). Quantitative analysis was assessed using Image J software. Data represent the fold change relative to that of shRNA control cells or empty vector control cells obtained from at least three independent experiments”.

Comments: Fig 3 a-c add PC-3 cells in Y axis as was done for DU-145 in Fig 3d

Response: Fig. 3a and 3b have been modified and are now including results from the 22Rv1 cells knockdown for CdGAP.

Comments: It is confusing to have data relative to fig 3 in supplementary fig 2 (d,e,f)

Response: These results are now shown in Fig.3g.

Comments: Supple Fig 2 e and f, it is not clear why it was assessed the adhesion to collagen and fibronectin. these are cell lines that adapt to plastic from passage one, and no difference of `adhesion` growth on plastic or ECM coated well is unsurprising, also when silencing CdGAP, that if was critical in adhesion would have had an effect also on plastic. The assay/s that would be more interesting would be the differences in phenotype, shape/protrusions/migration etc.

Response: We agree with the reviewer that we would probably have obtained similar results on plastic. That said, CdGAP has been shown to regulate integrin-dependent changes in cell motility and morphology (Lalonde et al, Curr. Biol. 2006), therefore this is the reason behind performing these assays on extracellular matrix proteins such as fibronectin and collagen. To assess differences between shCon cells and shCdGAP PC-3 cells, we have quantified the cell morphology changes showing a rounded cell morphology with a decreased cell area and cell aspect ratio in CdGAP-depleted PC-3 cells (Fig.2f). In addition, we have performed wound healing assays showing that CdGAP-depleted PC-3 cells were significantly less efficient to migrate in a wound-healing assay over a period of 27 hours (Fig. 3c and Supplementary Movie 1,2). Even though 22Rv1 cells were less migratory than PC-3 cells, loss of CdGAP in 22Rv1 cells significantly reduced the wound confluence compared to control cells (Fig. 3d and Supplementary Movie 3,4).

Comments: Fig 3 g The MTT assays measures the enzymatic activity rate. Proliferation assay by flow cytometry? If would be helpful to have the blot for CdGAP performed at 2,3,4,5 day post in parallel with the MTT assay.

Response: We have included the western blots of CdGAP over a period of 5 days in PC-3 and 22Rv1 cells in Supplementary Fig. 3f, showing that the expression of CdGAP remains abolished over the period of times performed for the MTT assays. We assessed the role of CdGAP on G1 cell cycle progression by flow cytometry analysis in Fig. 6c.

Comments: Line 167 rather than promoter, regulator, at this point of the story.

Response: It has been changed.

Comments: Line 231 delete migration and invasion as data in fig 5 do not show that.

Response: It has been modified.

Comments: Fig 4 it does not look you gained mechanistic inside with the data in this figure, because you do not show a list of possible genes affected resulting by the analisis in fig a-d, but you studied

genes known to be involved in breast cancer model. Furthermore, silencing CdGAP you do not know if these effect on Snail/slugs are direct or indirect.

Response: Fig. 4 is now Fig. 5. The differential gene expression analysis identified 1384 upregulated and 720 downregulated mRNAs in CdGAP-depleted PC-3 cells compared to control cells (Fig. 5a; list of genes provided as Supplementary Table 1). The analyses of the modified gene expression in Fig. 5e and f represent the validation of a subset of differentially expressed EMT genes. Similarly, Fig. 6a and b show a subset of differentially expressed genes encoding cell cycle checkpoint proteins. Indeed, we don't know if CdGAP interacts with the snail/slugs promoters, similar to the E-cadherin promoter, and this is of great interest and part of our future research endeavours.

Comments: Fig 6b The sentence in the legend is not clear. Are these tumour from 3 animals? What about the others?

Response: Fig. 6b is now Fig. 7b. We have modified the figure 7 legend accordingly on lines 923-924: "Representative photographs of endpoint tumors that formed in control (n=12) and shCdGAP (n=11) groups of mice."

Comments: In fig 6c are the data average of all 12 and 11 mice?

Response: Fig. 6c is now Fig. 7c. Yes. We have modified the figure legend on lines 924-926: **c** Growth curves of subcutaneously formed tumors. Tumor volume was measured three times a week up to 34 days and is presented as the mean volume of each group (control=12; shCdGAP=11). Error bars indicate standard deviation (SD).

Comments: Fig 7 J is very hard to understand, panel I too.

Response: Fig. 7 is now Fig. 8. We have clarified the Figure legend and the Figures 8J and I, highlighting in red CdGAP-T showing that high CdGAP expression in tumor (CdGAP-T) cores is a prognostic factor for progression to bone metastasis.

Comments: CdGAP-depleted pc3 cells:

Adhesion to plate: no difference (plastic/fn/collagen) (30 minutes)

MTT enzymatic activity for cell proliferation: decreases (5 days, time course/days)

COLONY formation, cell survival test/n divisions: decreases (10 days)

Ki67 nuclear stain for proliferation, no effect (mice inj with PC3)

How do you explain no proliferation *in vivo*?

Response: We provide explanations for these differences in the Discussion section on lines 322-334: "Further investigation of the proliferative capacities using *in vivo* subcutaneous injections demonstrated that CdGAP-depleted tumors exhibited delayed tumor onset, reduced tumor volume and tumor weight, in comparison to control tumors and this further substantiated the results obtained from the *in vitro* experiments. In contrast, prostate orthotopic injection of CdGAP-depleted cells did not alter the formation of primary tumors. These differences highlight the importance of the tumor microenvironment and stroma-tumor interaction in prostate cancer growth and progression. Cancer cells are sensitive to their surrounding cells and factors that contribute to reprogramming the tumor cells to either grow or arrest proliferation. The global transcriptional reprogramming in CdGAP-depleted PC-3 cells may support a positive niche for the tumors to develop in prostate tissue environment, which may be different in a subcutaneous tumor context. For instance, the upregulation of regulatory factors including TGF β and FGF1 in CdGAP-depleted cells could differentially influence the role of CdGAP in prostate cancer growth in a specific tumor microenvironment."

REVIEWERS' COMMENTS:

Reviewer #1 (Remarks to the Author):

The revised manuscript has been significantly improved and it is of major mechanistic significance. Insightful response by the authors.

Reviewer #2 (Remarks to the Author):

The authors replied to all comments and reshaped the manuscript to improve it substantially.